# TLR4 endocytosis and endosomal TLR4 signaling are distinct and independent outcomes of TLR4 activation

Thomas E Schultz [ID][1], Carmen D Mathmann [ID][1], Leslie C Domínguez Cadena [ID][1], Timothy W Muusse [ID][2], Hyoyoung Kim [ID][2], James W Wells[1], Glen C Ulett [ID][3], Jessica A Hamerman[4], Andrew J Brooks [ID][1], Bostjan Kobe [ID][2,5], Matthew J Sweet[5], Katryn J Stacey [ID][2] & Antje Blumenthal [ID][1✉]

## Abstract

Toll-like receptor 4 (TLR4) signaling at the plasma membrane and in endosomes results in distinct contributions to inflammation and host defence. Current understanding indicates that endocytosis of cell surface-activated TLR4 is required to enable subsequent signaling from endosomes. Contrary to this prevailing model, our data show that endosomal TLR4 signaling is not reliant on cell surface-expressed TLR4 or ligand-induced TLR4 endocytosis. Moreover, previously recognized requirements for the accessory molecule CD14 in TLR4 endocytosis and endosomal signaling are likely attributable to CD14 binding as well as trafficking and transferring lipopolysaccharide (LPS) to TLR4 at different subcellular localizations. TLR4 endocytosis requires the TLR4 intracellular signaling domain, contributions by phospholipase C gamma 2, spleen tyrosine kinase, E1/E2 ubiquitination enzymes, but not canonical TLR signaling adaptors and cascades. Thus, our study identifies independently operating TLR4 signaling modes that control TLR4 endocytosis, pro-inflammatory cell surface-derived, as well as endosomal TLR4 signaling. This revised understanding of how TLR4 functions within cells might be harnessed to selectively amplify or restrict TLR4 activation for the development of adjuvants, vaccines and therapeutics.

**Keywords** Macrophage; TLR4; Signaling; Endosome; LPS
**Subject Categories** Membranes & Trafficking; Microbiology, Virology & Host Pathogen Interaction; Signal Transduction

## Introduction

Immune recognition of microorganisms is governed by pattern recognition receptors (PRR), genetically encoded receptors that bind evolutionarily conserved microbial cellular components and host-derived danger signals (Barton and Medzhitov, 2002). Whereas PRR activation is critical for resolution of infection, dysregulation of PRR functions can be detrimental to the host (Takeuchi and Akira, 2010). Consequently, tight molecular regulation of PRR activation and signaling is essential for effective immune responses. Toll-like receptors (TLR) are PRRs that recognize a diverse range of microbial and host-derived molecules (Akira et al, 2006; Kawai et al, 2024). TLR4, best characterized for its recognition of lipopolysaccharide (LPS) from Gram-negative bacterial cell walls, is critical for initiating inflammation and effective bacterial clearance (Deng et al, 2013; Zhang et al, 2014). Yet, TLR4 hyper-activation contributes to pathology in inflammatory disorders (Ciesielska et al, 2021), highlighting the pathophysiological importance of understanding the molecular mechanisms of TLR4 activation and regulation.

As type I single-pass transmembrane proteins, TLRs consist of an extracellular leucine-rich repeat (LRR) domain engaged in ligand recognition, a transmembrane domain, and an intracellular Toll-Interleukin-1-Receptor (TIR) domain required for signal transduction (Botos et al, 2011) through recruitment of adaptor proteins including MyD88 (myeloid differentiation primary response gene 88), MAL (MyD88 adaptor-like), TRIF (TIR domain-containing adaptor-inducing interferon-β), TRAM (TRIF-related adaptor molecule) and BCAP (B cell adaptor for phosphoinositide 3-kinase) (Kenny and O'Neill, 2008). TLR4 initiates intracellular signaling from both the plasma membrane and endosomal compartments (Hoebe et al, 2003; Poltorak et al, 1998; Yamamoto et al, 2003). Ligand recognition by TLR4 results in the assembly of the multimeric myddosome complex (Motshwene et al, 2009), which drives activation of NF-κB and MAPK, and consequently expression of pro-inflammatory cytokines and antimicrobial defence mechanisms (Fitzgerald et al, 2001; Kawai et al, 2024). In contrast, endosomal TLR4 signaling has been positioned to TRAF3-positive endosomal compartments that enable recruitment of the adaptors TRAM and TRIF to TLR4, thereby facilitating activation of TBK-1 and IKKε signaling that

[1]Frazer Institute, The University of Queensland, Brisbane, QLD 4102, Australia. [2]School of Chemistry and Molecular Biosciences, The University of Queensland, Brisbane, QLD 4072, Australia. [3]School of Pharmacy and Medical Sciences and Menzies Health Institute Queensland, Griffith University, Gold Coast, QLD 4215, Australia. [4]Immunology Program, Benaroya Research Institute, Seattle, WA 98101, USA. [5]Institute for Molecular Bioscience, The University of Queensland, Brisbane, QLD 4072, Australia.
✉E-mail: a.blumenthal@uq.edu.au

results in expression of type I interferons (IFN) (Kagan et al, 2008; Tanimura et al, 2008). Endosomal TLR4 signaling also contributes to sustained NF-κB activation through RIPK1-dependent signalling (Cusson-Hermance et al, 2005; Sato et al, 2003). TLR4 endocytosis upon activation has been positioned as a key negative regulatory mechanism of MAL/MyD88-dependent pro-inflammatory TLR4 signaling (Husebye et al, 2006; Latz et al, 2002), as well as the necessary prerequisite for relocation of activated TLR4 to endosomal compartments to enable TRAM/TRIF-mediated TLR4 signaling (Kagan et al, 2008). Among TLRs, TLR4 appears to be unique in exhibiting these two spatially separated, discrete signaling mechanisms. This might offer opportunities for targeted engagement or inhibition of distinct TLR4 functions. Advancing such opportunities, however, requires detailed understanding of the molecular mechanisms underpinning and connecting TLR4 signaling modes.

Ligand-induced endocytosis is a common regulatory mechanism for cell surface receptor functions and recycling (Sorkin and von Zastrow, 2009). In the case of TLR4, receptor activity has been proposed to not control endocytosis, based on findings that macrophages expressing signaling-incompetent TLR4 (Tan et al, 2015; Zanoni et al, 2011) or lacking TLR signaling adaptors (Rajaiah et al, 2015; Zanoni et al, 2011) exhibited no defects in LPS-induced TLR4 internalization. Instead, the current model positions CD14, a GPI-anchored LPS-binding receptor, as the key driver of LPS-induced TLR4 endocytosis (Tan and Kagan, 2017; Tan et al, 2015; Zanoni et al, 2011). However, findings that synthetic TLR4 agonists induced TLR4 endocytosis and type I IFN expression in $Cd14^{-/-}$ macrophages (Rajaiah et al, 2015) indicated that CD14 is not essential for TLR4 endocytosis. Moreover, observations of impaired endocytosis of mutant TLR4 that exhibited compromised signaling ability (Kobayashi et al, 2006; Tsukamoto et al, 2018) suggest that TLR4 activity may, at least in part, contribute to TLR4 endocytosis. An active role of TLR4 in driving TLR4 endocytosis might also reposition the findings that MD-2, the TLR4 accessory molecule that accommodates LPS for TLR4 activation, directs TLR4 endocytosis (Tan et al, 2015). Thus, the current understanding of the hierarchy, interplay and identities of molecular mechanisms governing TLR4 endocytosis and the relay between TLR4 signaling outputs remains incomplete.

Here we show that endosomal TLR4 signaling and type I IFN expression in macrophages can occur independent of TLR4 surface expression and ligand-induced endocytosis. Through the use of CD14-dependent and -independent TLR4 agonists, our data indicate that CD14 contributions to TLR4 endocytosis and endosomal TLR4 signaling are consistent with CD14 binding and transferring LPS to TLR4, rather than independently controlling TLR4 endocytosis. Our data further show that TLR4 TIR domain signaling activity is essential for TLR4 endocytosis, independent of TLR signaling adaptors. Instead, TLR4 endocytosis was facilitated by phospholipase C gamma 2 (PLCγ2), spleen tyrosine kinase (SYK), and E1 and E2 ubiquitination enzyme activity outside of the canonical TLR4-MAL-MyD88 and TLR4-TRAM-TRIF intracellular signaling cascades. Our findings position TLR4 endocytosis as a consequence of TLR4 activity, resulting in degradation of endocytosed TLR4 and thereby restriction of pro-inflammatory TLR4 signaling at the cell surface. Moreover, these data identify that surface-derived and endosomal TLR4 signaling are independent, dissociable events not requiring TLR4 endocytosis as a molecular switch.

# Results

## TLR4 endocytosis is not a prerequisite for endosomal TLR4 signaling

The current notion that TLR4 endocytosis is a pre-requisite for endosomal TLR4 signaling arose from experiments where dynasore, a small molecule inhibitor of dynamin-1/-2 and mitochondrial dynamin Drp1 (Macia et al, 2006), inhibited the LPS-induced decline in macrophage TLR4 surface expression and *Ifnb1* mRNA expression (Kagan et al, 2008). Together with observations that TLR4, TRAM and TRAF3 localized to Rab5a-positive intracellular compartments in resting macrophages, it was proposed that dynamin-dependent translocation of TLR4 to early endosomes was required for TLR4-dependent type I IFN expression (Kagan et al, 2008). Consistent with this and other reports (Rajaiah et al, 2015), we found that dynasore abolished LPS-induced TLR4 endocytosis and significantly impaired *Ifnb1* expression in mouse bone marrow-derived macrophages (BMM) (Fig. EV1A,B). However, dynasore also significantly impaired LPS-induced cytokine responses attributable, at least in part, to TLR4-MAL-MyD88 signaling including *Il1b* expression (Fig. EV1B), aligning with previous observations related to IL-6 release (Kagan et al, 2008). Dynasore treatment also impaired cytokine mRNA expression in response to TLRs that signal from the cell surface (TLR2, TLR5) and endosomal compartments (TLR3, TLR9) (Fig. EV1C), indicating that dynasore impairs TLR functions beyond receptor internalization. We therefore re-visited interpretation of the impact of dynasore on TLR4 functions in light of findings on dynamin-independent effects of dynasore, in particular as a disruptor of lipid rafts (Park et al, 2013) (Preta et al, 2015). Lipid rafts are critical for activation of cell surface TLR4 (Latz et al, 2002; Triantafilou et al, 2001; Triantafilou et al, 2002). Thus, we queried how lipid raft disruption affected TLR4 endocytosis and cytokine responses in macrophages.

To address this, we used filipin and methyl-β-cyclodextrin (MβCD), which disrupt lipid rafts by binding and extracting cholesterol, respectively (Awasthi-Kalia et al, 2001). Filipin and MβCD abolished LPS-induced TLR4 endocytosis in primary BMM and immortalized BMMs (iBMM) (Fig. 1A,B; Appendix Fig. S1A), with some impact on TLR4 surface expression in resting macrophages (Appendix Fig. S1B). In the absence of TLR4 endocytosis, LPS-induced *Ifnb1* expression was not impaired but rather enhanced upon filipin or MβCD treatment (Fig. 1C; Appendix Fig. S1A), contrary to what would have been expected. Filipin or MβCD treatment significantly impaired LPS-induced *Il1b* expression in primary BMM but showed minor effects on *Il1b* expression in iBMM (Fig. 1D; Appendix Fig. S1A). Using inhibitors of TLR4 activity (TAK-242) (Takashima et al, 2009) and TBK1/IKKε (MRT67307) (Petherick et al, 2015) as well as TLR adaptor-deficient macrophages, we determined that endocytosis-independent *Ifnb1* expression was a bona fide product of TLR4-TRAM-TRIF signaling (Fig. EV1D; Appendix Fig. S2A,B). In contrast, maximal *Il1b* expression relied on both TLR4-MAL-MyD88 and TLR4-TRAM-TRIF signaling (Fig. EV1E; Appendix Fig. S2C), as described previously (Yamamoto et al, 2003). Consistent with intact *Ifnb1* expression, LPS-induced TBK1 phosphorylation was unaffected or even enhanced by lipid raft disruption (Fig. EV1F). Impaired TBK1 phosphorylation by dynasore treatment was consistent with previous reports (Kagan et al, 2008) and is likely reflective of the complex impact dynasore

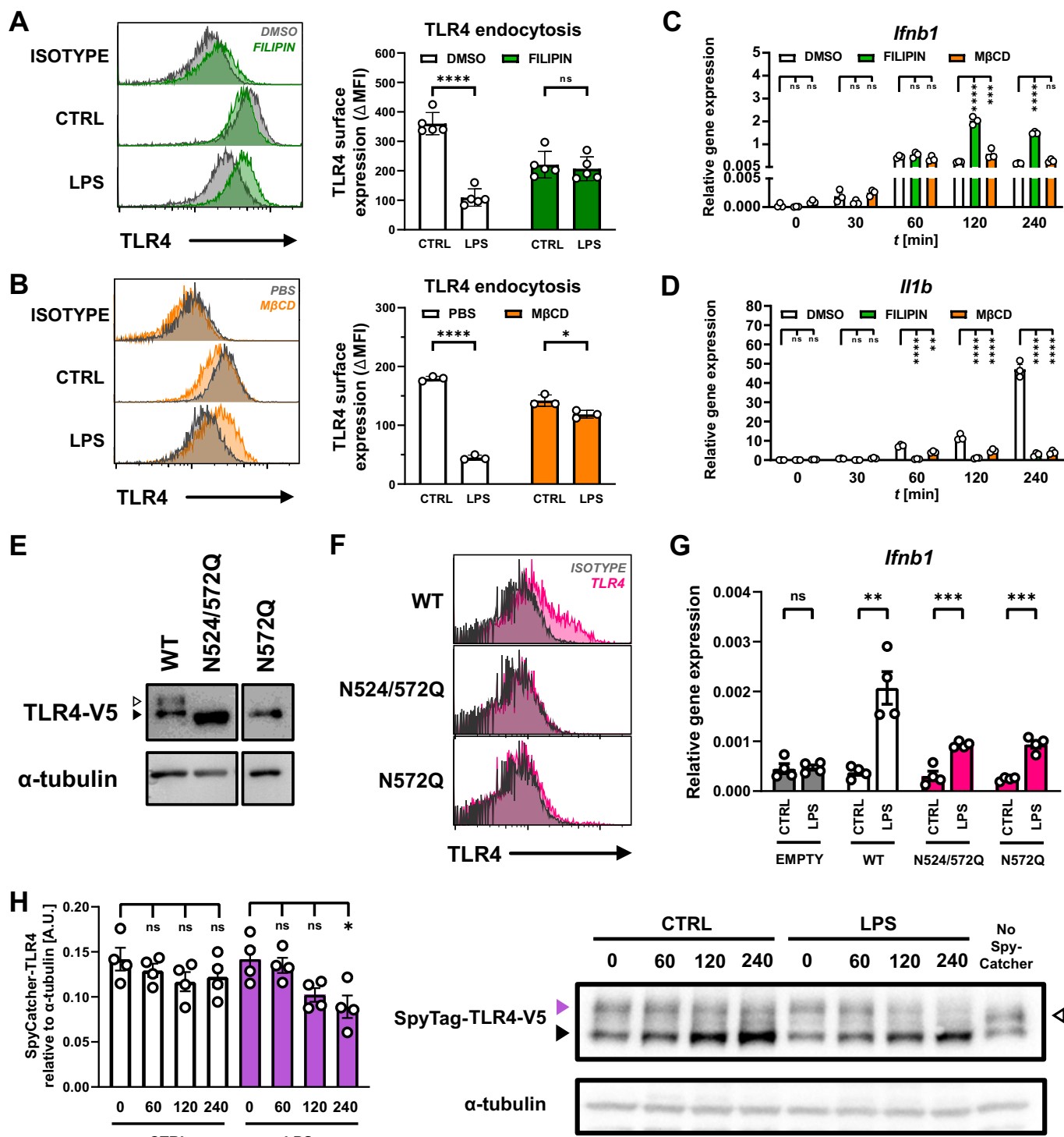

exerts on macrophages. These data revealed that endosomal TLR4 signaling can proceed in the absence of ligand-induced TLR4 endocytosis.

To further interrogate the relationship between cell-surface TLR4 and endosomal TLR4 signaling, we introduced mutations in N-linked glycosylation sites in the TLR4 extracellular domain that had previously been reported to abolish TLR4 expression at the plasma membrane (da

Silva Correia and Ulevitch, 2002). We generated TLR4-deficient RAW264.7 macrophages (RAW^TLR4ko) using CRISPR/Cas9 (Appendix Fig. S3A,B) and stably transfected these cells with expression vectors encoding either wild-type mouse TLR4 (mTLR4^WT) or mTLR4 mutated at glycosylation sites N524 and N572 to generate cells expressing mTLR4^N524/572Q or mTLR4^N572Q. Consistent with previous reports (da Silva Correia and Ulevitch, 2002; Husebye et al, 2006), our immunoblot analysis showed

**Figure 1.** **Expression and endocytosis of surface TLR4 are not required for LPS-induced TLR4 endosomal signaling.**

(A, B) TLR4 endocytosis was assessed in WT BMM treated for 60 min with lipid raft inhibitors (A) filipin (5 μM) or (B) MβCD (10 mM), followed by stimulation with LPS (100 EU/mL) for 120 min, or left unstimulated (CTRL). (C, D) Impact of lipid raft inhibitors filipin or MβCD on LPS-induced (C) *Ifnb1* and (D) *Il1b* mRNA expression in WT BMM stimulated with LPS (100 EU/mL) for the indicated times. (E) Immunoblots of TLR4-V5 in RAW$^{TLR4ko}$ cells stably reconstituted with mTLR4$^{WT}$, mTLR4$^{N524/572Q}$ or mTLR4$^{N572Q}$. Black arrow head indicates low-glycosylated intracellular TLR4, white arrow head indicates highly-glycosylated cell surface-expressed TLR4. (F) Flow cytometric analysis of TLR4 surface expression in RAW$^{TLR4ko}$ cells stably expressing mTLR4$^{WT}$, mTLR4$^{N524/572Q}$ or mTLR4$^{N572Q}$. (G) LPS-induced *Ifnb1* expression assessed by qRT-PCR in RAW$^{TLR4ko}$ cells stably expressing mTLR4$^{WT}$, mTLR4$^{N524/572Q}$ or mTLR4$^{N572Q}$. (H) Analysis of V5-tagged TLR4 via immunoblot in RAW$^{TLR4ko}$ cells expressing SpyTag-mTLR4$^{WT}$-V5 incubated with SpyCatcher protein, followed by stimulation with LPS (100 EU/mL), or left unstimulated, across indicated time points. Purple arrow head indicates SpyCatcher-labelled highly-glycosylated surface TLR4; black and white arrow heads indicate unlabelled intracellular and surface TLR4, respectively. Data information: Flow cytometry histograms and immunoblot images depict 1 representative of n = 3–5 biological replicates generated in independent experiments. Bar plots are mean ± SEM of n = 3–5 biological replicates generated in independent experiments indicated as data points. Unpaired two-tailed *t* test utilized in (A, B, G); ordinary two-way ANOVA with Dunnett's multiple comparison test utilized in (C, D). Ordinary one-way ANOVA with Dunnett's multiple comparison test utilized in (H). *P < 0.05; **P < 0.01; ***P < 0.001, ****P < 0.0001, ns = not significant. Source data are available online for this figure.

that RAW$^{TLR4ko}$ cells reconstituted with mTLR4$^{N524/572Q}$ or mTLR4$^{N572Q}$ lacked the highly-glycosylated cell surface-associated form of TLR4, while containing the minimally-glycosylated intracellular TLR4 (Fig. 1E). Despite lacking cell-surface expressed TLR4 (Fig. 1F), RAW$^{TLR4ko}$ cells expressing mTLR4$^{N524/572Q}$ or mTLR4$^{N572Q}$ displayed LPS-induced *Ifnb1* expression (Fig. 1G). These data demonstrated that macrophage responses attributed to TLR4 activation in endosomes can be initiated in the absence of cell surface-expressed TLR4.

Ligand-induced TLR4 endocytosis was originally described as a negative regulator of TLR4 activation through removal and degradation of cell surface-expressed TLR4 (Husebye et al, 2006). We revisited this using the SpyTag/SpyCatcher protein labelling system (Zakeri et al, 2012) to covalently label and track cell-surface TLR4 at the time of LPS stimulation. To this end, we used our RAW$^{TLR4ko}$ cells (Appendix Fig. S3A) to stably express mTLR4 N-terminally tagged with SpyTag (13 amino acids). The cells were then incubated with the SpyCatcher protein and stimulated with LPS. Immunoblot analyses confirmed an increase in the molecular weight of cell-surface expressed TLR4 corresponding to the binding of SpyCatcher (15 kDa), whereas the intracellular TLR4 pool not exposed to SpyCatcher protein showed no shift in molecular weight (Appendix Fig. S3C). LPS stimulation of SpyCatcher-labelled cells resulted in a significant decrease in the band corresponding to SpyCatcher-labelled TLR4, suggesting degradation of cell-surface exposed TLR4 upon LPS stimulation that substantially exceeds turnover of cell-surface TLR4 in resting cells (Fig. 1H). These findings align with a previous interpretation that ligand-induced TLR4 endocytosis curbs pro-inflammatory TLR4 signaling at the cell surface through the internalization and degradation of activated TLR4 (Husebye et al, 2006). In contrast, the pool of unlabelled intracellular TLR4 was maintained or even increased (Fig. 1H), the latter potentially representing replenishing of the cellular TLR4 pool after stimulation or a shift towards endosomal TLR4 signaling. We noted that SpyTag-TLR4 accumulated in unstimulated cells (with or without SpyCatcher labelling) (Fig. 1H; Appendix Fig. S3D), which might reflect a characteristic of this specific tagged TLR4 construct. Collectively, these data demonstrated that endosomal TLR4 signaling outputs occur in the absence of cell-surface expressed TLR4 and do not require ligand-induced TLR4 endocytosis.

## LPS-induced *Ifnb1* expression aligns with CD14 endocytosis

To further probe that TLR4 endocytosis and *Ifnb1* expression are not inextricably linked, we compared the impact of inhibitors targeting macropinocytosis (EIPA) (Koivusalo et al, 2010), clathrin-mediated endocytosis (Pitstop-2) (Rennick et al, 2021) and dynamin-dependent endocytosis (prochlorperazine/PCZ) (Chew et al, 2020) on LPS-induced TLR4 endocytosis, *Ifnb1* and *Il1b* expression. We initially confirmed that the small molecule inhibitors affected uptake of FITC-labeled dextran (70 kDa, macropinosome cargo) and transferrin (cargo for clathrin/dynamin-dependent endocytosis) in resting primary macrophages consistent with their respective targeted cellular processes (Appendix Fig. S4A,B). Aligning the impact of the endocytosis inhibitors on LPS-induced TLR4 endocytosis with cytokine expression, we observed more prominent overlap with inhibition patterns for *Il1b* rather than *Ifnb1* expression (Fig. 2A,B; Appendix Fig. S4C), lending further credence to the disconnect between TLR4 endocytosis and *Ifnb1* expression. We therefore considered the possibility that LPS might reach endosomal TLR4 independent of the internalization of cell surface-expressed TLR4. CD14 was originally described as an LPS uptake receptor (Wright et al, 1990) and was recently reported to support non-canonical inflammasome activation by shuttling LPS into the cytosol via endosomes (Kumari et al, 2023; Vasudevan et al, 2022). LPS-inducible *Ifnb1* expression requires CD14 (Jiang et al, 2005) (Appendix Fig. S4D) and we found that this requirement remained even when TLR4 endocytosis was abrogated by filipin treatment (Appendix Fig. S2A,B). Thus, we questioned whether perturbations of LPS-induced CD14 endocytosis aligned with *Ifnb1* expression in macrophages. Consistent with previous reports (Rajaiah et al, 2015; Schappe et al, 2018; Tan et al, 2015), LPS dose-dependently induced a decline in cell surface expression of CD14 (Appendix Fig. S4E). Endocytosis inhibitors affected CD14 endocytosis in patterns comparable to their impact on *Ifnb1* expression (Fig. 2A,B; Appendix Fig. S4F). Notably, neither disruptors of lipid rafts (filipin, MβCD) nor inhibition of TLR4 activity (TAK-242) impaired LPS-induced CD14 endocytosis (Appendix Fig. S4F), suggesting that CD14 can shuttle LPS into the cell independent of TLR4 activity and lipid raft integrity.

To further probe to requirements for CD14 for endosomal TLR4 activation we used the CD14-independent synthetic TLR4/MD-2 agonist 1Z105 (Hayashi et al, 2014). We initially confirmed that macrophage stimulation with 1Z105 induced TLR4 endocytosis comparably to LPS, as well as concomitant expression of *Il1b* and *Ifnb1* in a manner dependent on TLR4 but not CD14 (Appendix Fig. S5A–C). Filipin blocked 1Z105-induced TLR4 endocytosis and *Il1b* expression, but not *Ifnb1* expression (Fig. 2C,D), similar to our observations in LPS-stimulated macrophages. As with LPS stimulation, the impact of endocytosis inhibitors on 1Z105-induced TLR4 endocytosis did not mirror their impact on *Ifnb1*

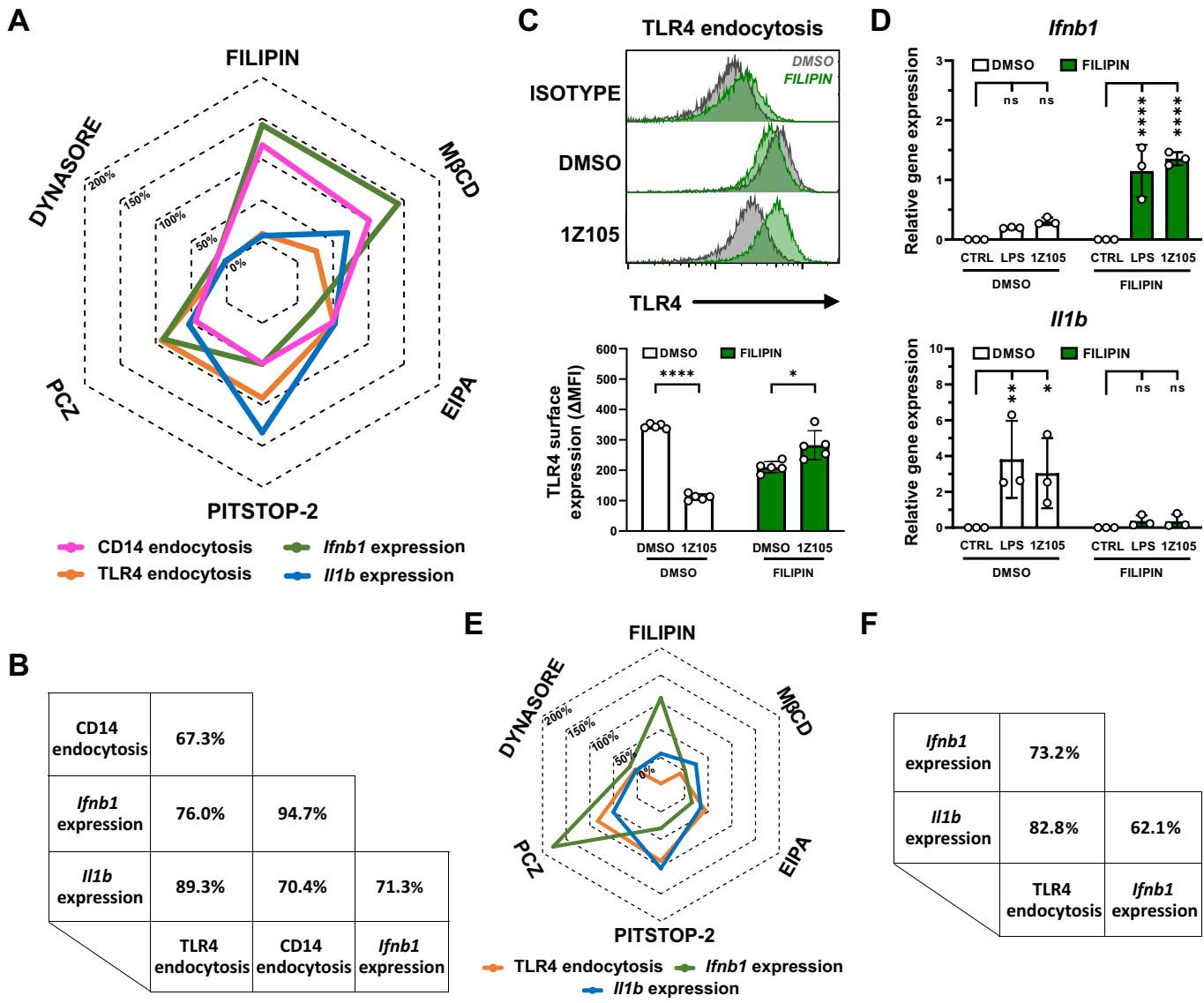

**Figure 2. Endocytosis of CD14 correlates with TLR4-mediated *Ifnb1* expression.**

(A, B) Comparison of impact by endocytosis inhibitors and lipid raft disruptors (filipin, MβCD, EIPA, Pitstop-2, PCZ, dynasore) on LPS-induced TLR4 endocytosis, *Ifnb1* expression, *Il1b* expression (100 EU/mL LPS) and CD14 endocytosis (10000 EU/mL LPS) in WT BMMs. Endocytosis and mRNA expression data in each independent experiment were expressed as a percentage of the parameter detected in cells treated with solvent control. (B) Percentage overlap of inhibitor perturbation profiles depicted in (A). (C, D) Impact of filipin treatment on (C) TLR4 endocytosis and (D) *Ifnb1* and *Il1b* expression induced by 1Z105 (10 μM) in WT BMMs. (E, F) Comparison of impact by endocytosis inhibitors and lipid raft disruptors on 1Z105-induced TLR4 endocytosis, *Ifnb1* expression and *Il1b* expression. Endocytosis and mRNA expression data in each independent experiment were expressed as a percentage of the parameter detected in cells treated with solvent control. (F) Percentage overlap of inhibitor perturbation profiles depicted in (E). Data information: Flow cytometry histograms depict 1 representative of $n = 5$ biological replicates generated in independent experiments. Bar plots are mean ± SEM of $n = 3–5$ biological replicates generated in independent experiments indicated as data points. Radar plots show means of $n = 3–5$ biological replicates generated in independent experiments. Unpaired two-tailed $t$ test utilized in (C). Ordinary two-way ANOVA with Sidak's multiple comparisons test utilized in (D). *$P < 0.05$; **$P < 0.01$; ****$P < 0.0001$, ns = not significant. Source data are available online for this figure.

expression (Fig. 2E,F; Appendix Fig. S4G,H) further highlighting dissociation between TLR4 endocytosis and endosomal TLR4 signaling. In contrast to LPS, the CD14-independent TLR4 agonist 1Z105 did not impact CD14 surface expression (Appendix Fig. S4E,F). Collectively, and together with the current knowledge on CD14 functions, these data indicate that TLR4 endocytosis per se does not require CD14. We propose that in the case of LPS recognition, CD14 endocytosis is ligand-induced (independent of

TLR4 activation) and transports LPS to endosomes resulting in activation of resident endosomal TLR4, independent of the activation and endocytosis of cell surface TLR4. Positioning CD14 as the shuttling receptor for LPS transfer to endosomal TLR4 (rather than directing TLR4 endocytosis), offers a model that reconciles the divergent requirements for CD14 in LPS- and 1Z105-induced *Ifnb1* expression (Rajaiah et al, 2015; Zanoni et al, 2011).

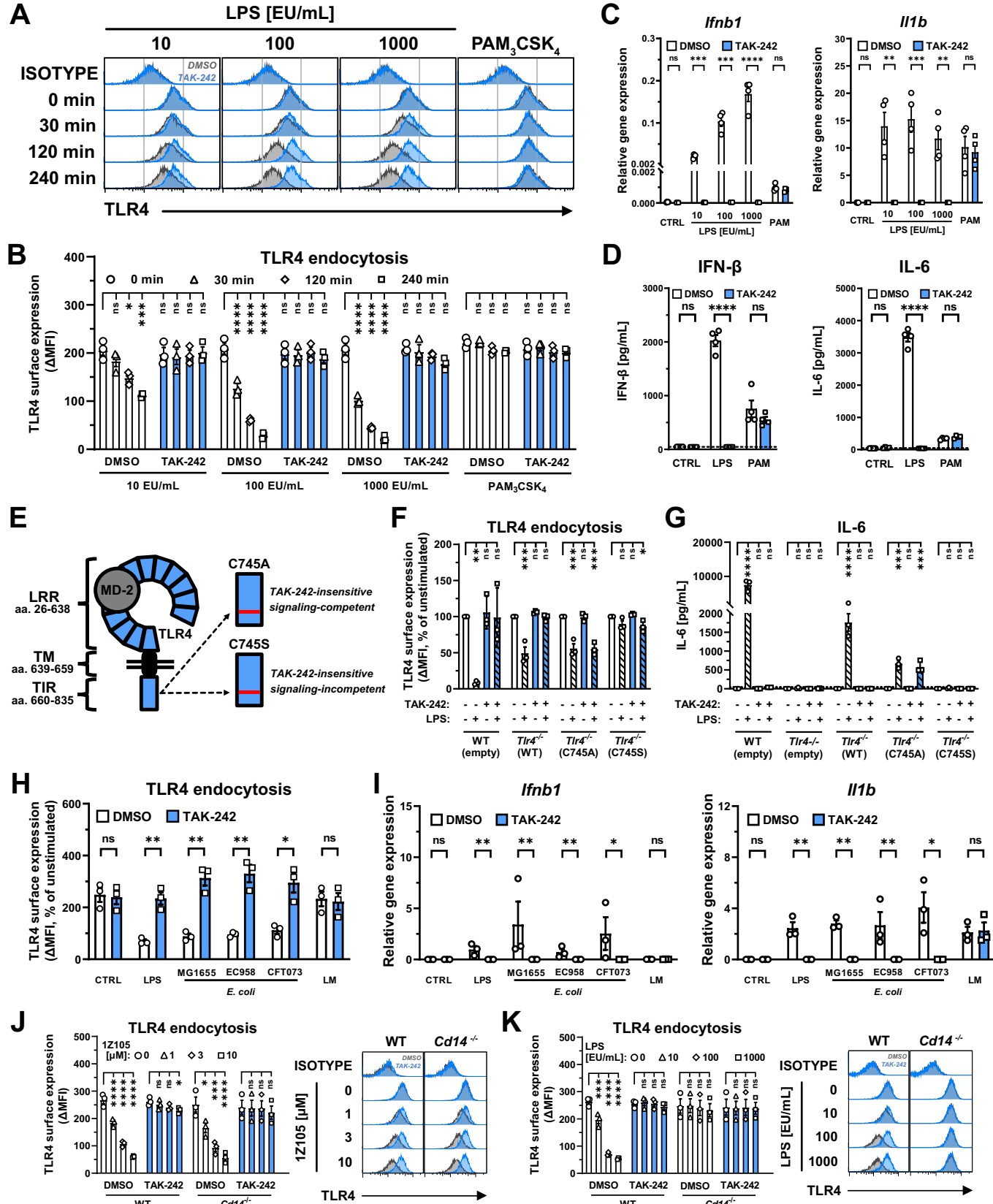

◀

**Figure 3.  TLR4 activity is required for ligand-induced TLR4 endocytosis.**

Impact of TLR4 inhibitor TAK-242 on (**A, B**) TLR4 surface expression at indicated time points, and cytokine (**C**) mRNA expression (1.5 h/90 min) and (**D**) protein secretion (4 h) in WT BMMs stimulated with LPS (10, 100, 1000 EU/mL), Pam$_3$CSK$_4$ (10 ng/mL), or left unstimulated (CTRL). (**E–G**) Retroviral reconstitution of *Tlr4*$^{−/−}$ BMMs with (**E**) mTLR4$^{WT}$, mTLR4$^{C745A}$, mTLR4$^{C745S}$ or empty vector were treated with TAK-242 for 1 h prior to LPS stimulation (100 EU/mL). (**F**) TLR4 surface expression at 4 h post-stimulation; (**G**) IL-6 release at 24 h post-stimulation. (**H, I**) Impact of TAK-242 on (**H**) TLR4 surface expression (120 min) or (**I**) *Ifnb1/Il1b* expression (90 min) in WT BMM infected with *E. coli* (strains MG1655, EC958, CFT073) or *L. monocytogenes* strain 10403S (LM) (MOI 10). (**J, K**) Impact of TAK-242 on TLR4 endocytosis of WT and *Cd14*$^{−/−}$ BMM in response to (**J**) CD14-independent TLR4 agonist 1Z105 (10 µM) or (**K**) CD14-dependent TLR4 agonist LPS (100 EU/mL). Data information: Flow cytometry histograms depict 1 representative of $n = 3$ biological replicates. Bar plots are mean ± SEM of $n = 3–4$ biological replicates generated in independent experiments indicated as data points. Ordinary two-way ANOVA with Dunnett's multiple comparisons test utilized in (**B, J, K**). Unpaired two-tailed *t* test utilized in (**C, D, H, I**). Ordinary one-way ANOVA with Dunnett's multiple comparisons test utilized in (**F, G**). *$P < 0.05$; **$P < 0.01$; ***$P < 0.001$, ****$P < 0.0001$, ns = not significant. Source data are available online for this figure.

## TLR4 endocytosis requires a signaling-competent TLR4 TIR domain

With a revised view that CD14 does not direct TLR4 endocytosis, we revisited whether TLR4 activation was required for TLR4 endocytosis. Previous studies addressing this question used TLR4 mutations (Kobayashi et al, 2006; Rajaiah et al, 2015; Tsukamoto et al, 2018; Zanoni et al, 2011) and activation with CD14-independent TLR4 agonists (Rajaiah et al, 2015), but a consistent model for the requirement of TLR4 activity has not arisen. To gain further insight into whether TLR4 activity directs TLR4 endocytosis, we initially used the TLR4 inhibitor TAK-242, which prevented LPS-induced TLR4 endocytosis in primary macrophages across time and a range of LPS doses (Fig. 3A,B), while not impacting basal TLR4 expression in resting cells (Appendix Fig. S6A). Parallel analyses of cytokine mRNA expression and protein secretion confirmed that TAK-242 blocked TLR4-MAL-MyD88- and TLR4-TRAM-TRIF-dependent signaling (Fig. 3C,D). TAK-242 selectively and covalently binds to a cysteine in the TLR4 TIR domain (C747 in human (h)TLR4), but not other TLRs or TLR adaptor proteins (Matsunaga et al, 2011; Takashima et al, 2009). Mutation of C747 in hTLR4 to serine rendered TLR4 unresponsive to LPS (Núñez Miguel et al, 2007; Shirey et al, 2020; Takashima et al, 2009), whereas mutation to alanine retained a degree of LPS responsiveness that was insensitive to TAK-242 treatment (Takashima et al, 2009). We confirmed the TLR4-specificity of TAK-242-mediated abrogation of LPS-induced TLR4 endocytosis by reconstituting primary *Tlr4*$^{−/−}$ BMM with retroviral vectors encoding either wild-type mouse TLR4 (mTLR4$^{WT}$) or mTLR4 bearing the equivalent mutations of the corresponding cysteine in mTLR4, mTLR4$^{C745S}$ or mTLR4$^{C745A}$ (Fig. 3E). Surface mTLR4 expression in reconstituted cells was comparable between mTLR4$^{WT}$, mTLR4$^{C745S}$, and mTLR4$^{C745A}$ (Appendix Fig. S6B–D). Surface expression of signaling-incompetent mTLR4$^{C745S}$ did not decline in response to LPS stimulation and was not affected by TAK-242. In contrast, mTLR4$^{C745A}$ surface expression was reduced upon LPS stimulation, yet in a manner insensitive to TAK-242 (Fig. 3F; Appendix Fig. S6C,D). LPS-induced IL-6 and CXCL10 production reflected the relative signaling capacity and TAK-242 sensitivity of wild-type and mutant mTLR4 (Fig. 3G; Appendix Fig. S6E). Thus, the ablation of TLR4 endocytosis by TAK-242 aligned with presence of the TAK-242 binding site in the TLR4 TIR domain as well as TLR4 signaling capacity. We confirmed requirement of TLR4 activity for TLR4 endocytosis and cytokine expression in the context of infection with *E. coli* expressing rough (strain MG1655 (Rendueles et al, 2014)) and smooth LPS (clinical isolates CFT073 (Brzuszkiewicz et al, 2006), EC958 (Phan et al, 2013)) (Fig. 3H,I), suggesting that TLR4-mediated

TLR4 endocytosis occurs in the presence or absence of LPS O-antigen. Moreover, TAK-242 inhibited ligand-induced TLR4 endocytosis in response to both CD14-dependent LPS as well as CD14-independent 1Z105 (Fig. 3J,K).

To further corroborate the requirement for TLR4 signaling competency in TLR4 internalization, we assessed TLR4 endocytosis in macrophages carrying TLR4 with a proline-to-histidine mutation (P712H) in the TLR4 TIR domain. This mutation occurs in the C3H/HeJ mouse strain that shows severely impaired responsiveness to LPS (Poltorak et al, 1998). Using macrophages from C3H/HeJ mice (compared to C3H/HeN wild-type) as well as *Tlr4*$^{−/−}$ macrophages retrovirally reconstituted with mTLR4$^{P712H}$ (compared to mTLR4$^{WT}$), we found that TLR4 endocytosis and cytokine expression were greatly reduced across a dose range of LPS (Fig. 4A–D; Appendix Fig. S6F–J). Notably, a small level of TLR4 endocytosis was retained in C3H/HeJ BMM at high LPS concentration, consistent with previous observations (Tsukamoto et al, 2018) (Fig. 4A; Appendix Fig. S6G,H). This residual LPS-induced decline in TLR4 surface expression in C3H/HeJ macrophages was blocked by TAK-242 (Fig. 4A), suggesting that residual TLR4 activity controlling TLR4 endocytosis is retained within mTLR4 that carries the P712H mutation. Previous studies reported intact (Tan et al, 2015) or impaired TLR4 internalization (Kobayashi et al, 2006) upon LPS stimulation when the TLR4 TIR domain was truncated. In our hands, *Tlr4*$^{−/−}$ primary macrophages reconstituted with mTLR4 lacking a TIR domain (mTLR4$^{ΔTIR}$) (Fig. 4B) remained unresponsive to LPS both in terms of LPS-induced TLR4 endocytosis and cytokine expression (Fig. 4C,D; Appendix Fig. S6G,H). We independently confirmed the requirement of the TLR4 TIR domain for LPS-induced TLR4 endocytosis using RAW$^{TLR4ko}$ cells stably expressing mTLR4$^{ΔTIR}$ (Appendix Fig. S6K). Collectively, these data arising from genetic and pharmacologic approaches demonstrate that TLR4 activity and an intact TLR4 TIR domain are required for ligand-induced TLR4 endocytosis in macrophages. Such requirement of TLR4 activation for TLR4 endocytosis is consistent with lipid raft disruption preventing TLR4 endocytosis (Fig. 1) and accommodates the previously noted critical roles for both the LPS transfer receptor CD14 (Zanoni et al, 2011) and the TLR4 accessory molecule MD-2 (Tan et al, 2015) in the case of LPS-induced TLR4 endocytosis.

## TLR4 TIR domain-dependent endocytosis occurs independently of canonical TLR4 signaling pathways

Our findings that TLR4 TIR domain and signaling capacity are required for promotion of TLR4 endocytosis seem to contrast that

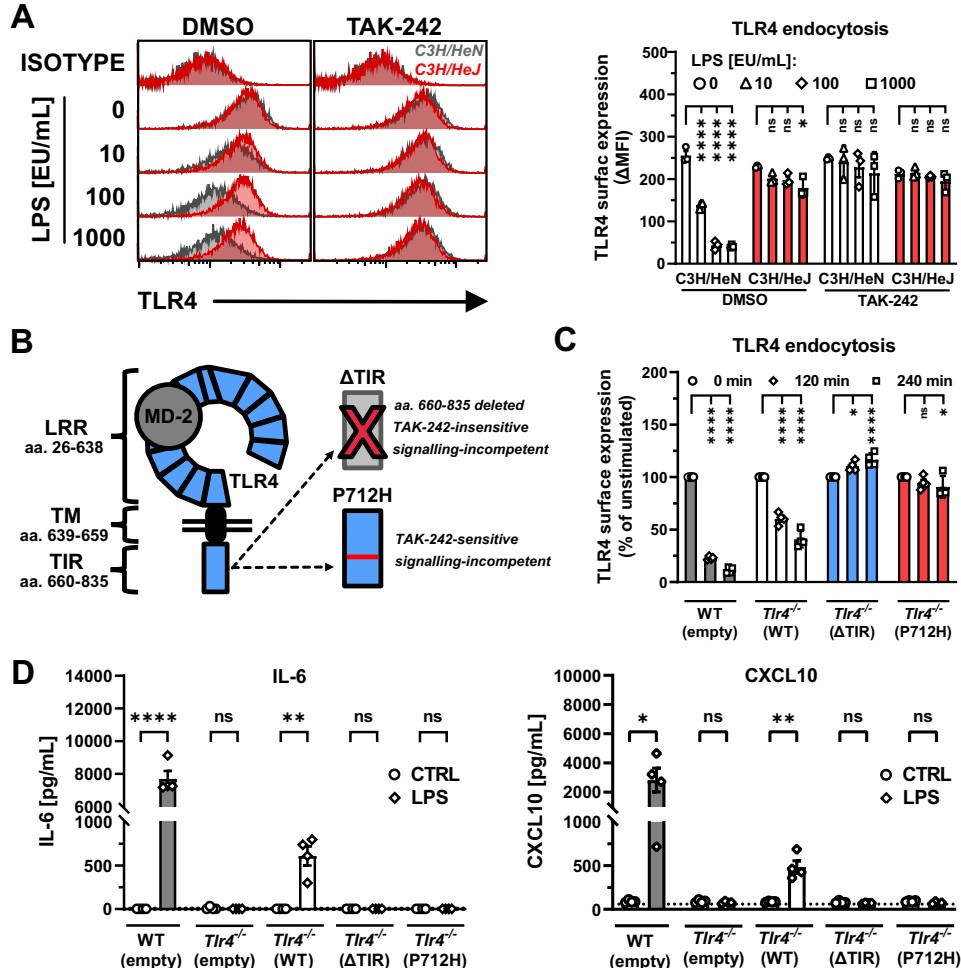

**Figure 4. Presence and signaling capability of the TLR4 TIR domain are prerequisites for ligand-induced TLR4 endocytosis.**

(A) Impact of TAK-242 treatment on TLR4 surface expression in C3H/HeN and C3H/HeJ BMM after LPS stimulation (indicated concentrations, 120 min). (B–D) $Tlr4^{-/-}$ BMMs retrovirally reconstituted with (B) mTLR4$^{WT}$, mTLR4$^{\Delta TIR}$, mTLR4$^{P712H}$ or empty vector were stimulated with LPS (100 EU/mL) and (C) TLR4 surface expression (0, 120, 240 min) and (D) IL-6 and CXCL10 concentrations (24 h) assessed. Data information: Flow cytometry histograms depict 1 representative of $n = 3$ biological replicates. Bar plots are mean ± SEM of $n = 3$–4 biological replicates generated in independent experiments indicated as data points. Ordinary two-way ANOVA with Dunnett's multiple comparisons test utilized in (A, C). Unpaired two-tailed $t$ test utilized in (D). $*P < 0.05$; $**P < 0.01$; $***P < 0.001$, $****P < 0.0001$, ns = not significant. Source data are available online for this figure.

TLR4 endocytosis occurs independent of the TLR adaptors MAL, MyD88, TRAM and TRIF as indicated in previous reports (Rajaiah et al, 2015; Zanoni et al, 2011) and our data (Appendix Fig. S2A). Our data further show that LPS-induced TLR4 endocytosis in macrophages deficient for MyD88, MAL, TRIF or TRAM was abrogated by TAK-242 (Fig. 5A,B; Appendix Fig. S7A), whereas cytokine expression required TLR4 activity and adaptors consistent with current understanding of TLR4 signaling pathway contributions (Appendix Fig. S7B–D). We made similar observations for the TLR4 adaptor BCAP (Ni et al, 2012; Troutman et al, 2012), which had not been previously assessed for contributions to TLR4 endocytosis (Fig. 5C; Appendix Fig. S7E). Consistent with a lack of TLR adaptor contributions to TLR4 endocytosis, inhibition of key components of the MAPK and TBK-1/IKKe signaling pathways did not affect LPS- and 1Z105-induced TLR4 endocytosis (Fig. 5D,E; Appendix Fig. S8A–D), while exhibiting expected impacts on kinase activation and cytokine expression (Appendix

Fig. S8E–G). The NF-κB signaling inhibitor SC-514 did not affect ligand-induced TLR4 endocytosis (Fig. 5F). In contrast, the NF-κB signaling inhibitor IKK-16 impaired TLR4 endocytosis to a small extent at 120 and 240 min post stimulation (Fig. 5D–F), which might reflect contributions of additional IKK-16 targets outside of canonical TLR4-NF-κB signaling (Hermanson et al, 2012). In summary, TLR4-mediated TLR4 endocytosis occurs independent of TLR signaling adaptors and their known downstream canonical TLR signaling cascades.

## Ubiquitination enzyme activity is required for ligand-induced TLR4 endocytosis

Functions of TLRs and components of TLR-driven intracellular signaling cascades are tightly regulated through the concerted impact of post-translational modifications, such as ubiquitination, to regulate protein function and stability (Hu and Sun, 2016). We

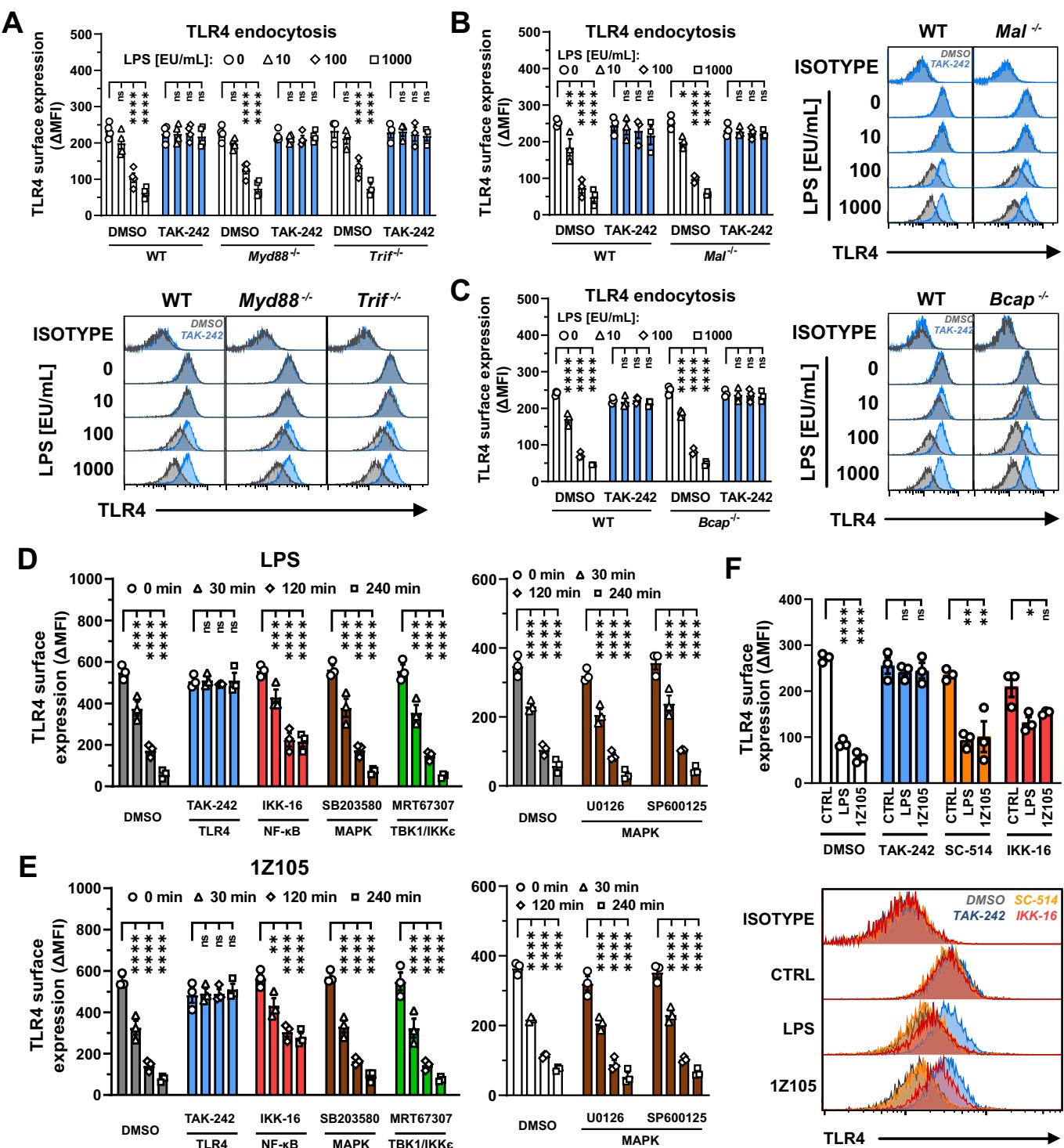

therefore reasoned that the TLR4-mediated process that facilitates TLR4 endocytosis outside of known TLR signaling mechanisms might be regulated by ubiquitination events. Covalent attachment of ubiquitin to target proteins requires a molecular cascade with contributions from three tiers of enzymes: E1 (ubiquitin-activating), E2 (ubiquitin-conjugating) and E3 (ubiquitin-ligating) (Middleton and Day, 2023). With 2 E1, ~40 E2, and, >600 E3

enzymes known in humans and mice, we prioritized small molecule inhibitors of E1 (PYR-41 (Yang et al, 2007), MLN7243 (Hyer et al, 2018)) and E2 (NSC697923 (Pulvino et al, 2012), BAY 11-7082 (Strickson et al, 2013)) enzymes after confirming that they did not affect TLR4 surface expression in resting cells and exhibited expected functional impact on macrophage cytokine responses (Appendix Fig. S9A,B) (Nakamura et al, 2022; Zinngrebe

**Figure 5. TLR4 TIR domain-dependent endocytosis occurs independent of canonical TLR signaling pathways.**

(A–C) Impact of adaptor protein deficiency on TAK-242-sensitive TLR4 endocytosis. WT and (A) $Myd88^{-/-}$ and $Trif^{-/-}$, (B) $Mal^{-/-}$ or (C) $Bcap^{-/-}$ BMM were treated with TAK-242 for 60 min prior to LPS stimulation (100 EU/mL) for 120 min. (D, E) Impact of inhibitors of canonical TLR signaling pathway components on TLR4 endocytosis. WT BMM pre-treated for 60 min with DMSO, TAK-242, IKK-16 (10 μM), SB203580 (10 μM), MRT67307 (5 μM), U0126 (10 μM) or SP600125 (10 μM) followed by stimulation with (D) LPS (100 EU/mL) or (E) 1Z105 (10 μM) over a time-course. (F) Impact of NF-κB pathway inhibitors IKK-16 or SC-514 on TLR4 endocytosis. WT BMM were pre-treated for 60 min with DMSO, TAK-242, IKK-16 or SC-514 (20 μM), followed by stimulation with LPS (100 EU/mL) for 120 min. Data information: Flow cytometry histograms depict 1 representative of $n = 3$ biological replicates. Bar plots are mean ± SEM of $n = 3$ biological replicates generated in independent experiments indicated as data points. Ordinary two-way ANOVA with Dunnett's multiple comparisons test utilized in (A–E). Ordinary one-way ANOVA with Dunnett's multiple comparisons test utilized in (F). $*P < 0.05$; $**P < 0.01$; $***P < 0.001$, $****P < 0.0001$, ns = not significant. Source data are available online for this figure.

et al, 2014; Zou and Zhang, 2021). Our data showed that inhibitors of E1 and E2 enzymes impaired or completely inhibited LPS- and 1Z105-induced TLR4 endocytosis (Fig. 6A–D). We noted that E1 and E2 enzyme inhibitors also blocked LPS-induced CD14 endocytosis, while exhibiting variable effects on general cellular endocytic uptake mechanisms (Appendix Fig. S9C,D). In contrast, inhibitors of enzymes mediating post-translational protein modification by NEDDylation (MLN4924 (Soucy et al, 2009)) and SUMOylation (TAK-981 (Langston et al, 2021)) did not affect LPS- or 1Z105-induced TLR4 endocytosis (Fig. 6C,D; Appendix Fig. S9E,F). Key regulatory mechanisms controlled by ubiquitination within canonical TLR4 signaling cascades lead to degradation of ubiquitinated proteins (e.g. IκB (Kanarek et al, 2010)). Yet, inhibition of proteasome activity (MG132 (Fiedler et al, 1998)) did not recapitulate the impact of E1 and E2 inhibitors. Instead, MG132 exhibited minor inhibition of TLR4 endocytosis reminiscent of the impact of IKK-16 (Figs. 5D–F and 6A,B). Thus, ubiquitination-mediated mechanisms appear to promote TLR4 endocytosis through pathways other than proteasomal degradation.

Ubiquitination regulates endocytosis of receptor tyrosine kinases (e.g. epidermal growth factor receptor (Haglund and Dikic, 2012)) and TLR4 ubiquitination has been reported (Abe et al, 2013; Bachmaier et al, 2007; Husebye et al, 2006). Canonical ubiquitination occurs on lysine residues (Malynn and Ma, 2010) of which the mTLR4 TIR domain contains 9. We questioned whether the requirement for the TLR4 TIR domain in governing TLR4 endocytosis reflected lysine ubiquitination of the TLR4 TIR domain. To address this, we utilized retroviral reconstitution of $Tlr4^{-/-}$ BMM with mTLR4^WT and mTLR4 bearing arginine substitutions for lysine residues (K663R/K664R/K692R/K727R/K730R/K771R/K774R/K810R/K817R compound mutations; mTLR4^9KΔR [all lysine residues in TIR domain mutated], mTLR4^7KΔR and mTLR4^5KΔR; Fig. EV2A) to examine the impact on TLR4 endocytosis. TLR4 lysine mutants exhibited surface expression comparable to mTLR4^WT, no defects in LPS-induced TLR4 endocytosis, and produced IL-6 and the interferon-inducible CXCL10 in response to LPS, albeit with somewhat reduced capacity in the case of IL-6 (Fig. EV2B,C). To assess impact of lysine mutations on cellular TLR4 protein pools, we used our RAW^TLR4ko cells (Appendix Fig. S3A,B) to stably express mTLR4^9KΔR. The higher molecular weight cell surface as well as the lower molecular weight intracellular forms of mTLR4^9KΔR were expressed at higher levels than mTLR4^WT (Fig. EV2D,E). These data indicated that TLR4 TIR domain lysine ubiquitination does not promote TLR4 endocytosis, but that TIR domain lysine residues might govern TLR4 expression levels, stability and/or turnover in macrophages.

Collectively, our observations, in combination with previous studies, suggest that TLR4 endocytosis is governed by a ubiquitin-dependent molecular process that is dependent on both TIR domain presence and signaling competence, but independent of adaptor proteins and canonical signaling.

## PLCγ2 promotes TLR4 endocytosis and type I IFN expression, and is stabilised downstream of TLR4

Previous studies implicated PLCγ2 (Chiang et al, 2012; Rajaiah et al, 2015; Zanoni et al, 2011), SYK (Rajaiah et al, 2015; Zanoni et al, 2011), and PI3K p110δ (Aksoy et al, 2012) as positive regulators of TLR4 endocytosis and endosomal TLR4 signaling outputs. We therefore asked how these regulators would be positioned in a revised model where TLR4 endocytosis was dissociated from endosomal TLR4 signaling. CRISPR/Cas9-mediated depletion of SYK significantly impaired LPS-induced TLR4 endocytosis in primary macrophages (Appendix Fig. S10A,B), supporting previous reports that SYK is a positive regulator of ligand-induced TLR4 endocytosis (Rajaiah et al, 2015; Zanoni et al, 2011). We sought to extend our studies using three small molecule inhibitors of SYK activity (R406, BAY 61-3606, piceatannol). However, these failed to impact LPS- or 1Z105-induced TLR4 endocytosis in macrophages and inhibited cytokine responses to varying degrees with no consistent pattern (Appendix Fig. S10C,D). While these tools were utilized in previous studies (Rajaiah et al, 2015; Zanoni et al, 2011), the reasons for divergent results between these and our experiments are not immediately obvious.

CRISPR/Cas9-mediated depletion of PI3K p110δ or inhibition with the PI3K p110δ-selective inhibitor CAL-101 did not significantly affect ligand-induced TLR4 endocytosis (Appendix Fig. S10E–G), suggesting that in the present system PI3K p110δ did not contribute to ligand-induced TLR4 endocytosis. While our findings contrast a previous report (Aksoy et al, 2012), elevated TLR4 surface expression in PI3K p110δ depleted cells might indicate a role for PI3K p110δ in TLR4 expression and turnover (Appendix Fig. S10F).

Nonetheless, it was noteworthy that in our experiments the inhibitor R406 abrogated $Ifnb1$ (but not $Il1b$) expression, and this was associated with inhibition of CD14 endocytosis (Appendix Fig. S10H). In addition, CAL-101 inhibited LPS-induced CD14 endocytosis and $Ifnb1$ (but not $Il1b$) expression to some extent (Appendix Fig. S10I,J). These data align with a model of TLR4 endocytosis being dissociated from endosomal TLR4 signaling, and might suggest contributions by SYK and PI3K p110δ activity to LPS-induced CD14 endocytosis.

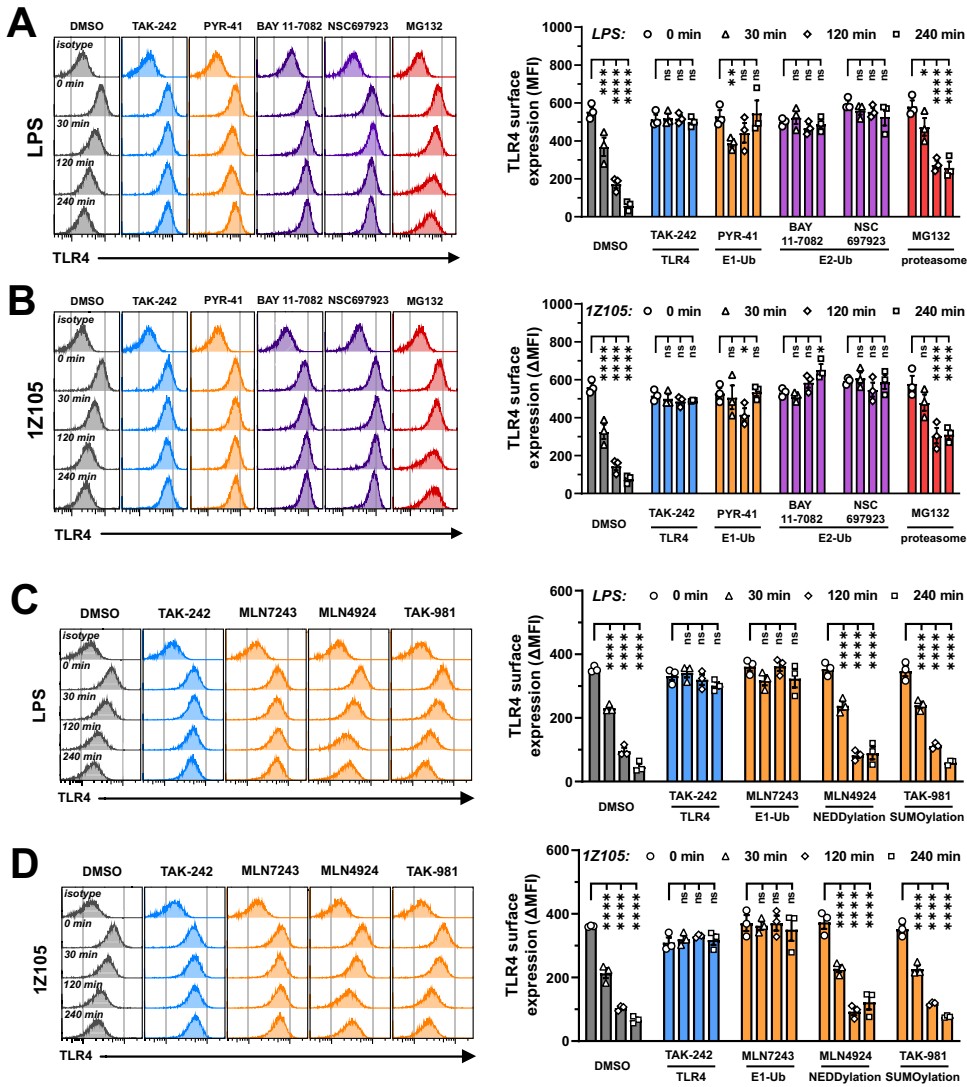

**Figure 6. A ubiquitination-linked cellular process, independent of canonical TLR signaling, governs TLR4 endocytosis.**

(A, B) Impact of ubiquitination enzyme inhibitors on TLR4 endocytosis. WT BMM were treated for 60 min with TAK-242, E1 inhibitor PYR-41 (25 μM), E2 inhibitors BAY 11-7082 (10 μM), NSC697923 (10 μM), or proteasome inhibitor MG132 (25 μM) followed by stimulation with (A) LPS (100 EU/mL) or (B) 1Z105 (10 μM) over a time-course. (C, D) Impact of NEDDylation and SUMOylation inhibitors on TLR4 endocytosis. WT BMM were treated for 60 min with TAK-242, MLN7243 (1 μM), MLN4924 (2.5 μM) or TAK-981 (2 μM) followed by stimulation with (C) LPS (100 EU/mL) or (D) 1Z105 (10 μM) over a time-course. Data information: Flow cytometry histograms depict 1 representative of $n = 3$ biological replicates. Bar plots are mean ± SEM of $n = 3$ biological replicates generated in independent experiments indicated as data points. Ordinary two-way ANOVA with Dunnett's multiple comparisons test utilized in (A–D). *$P < 0.05$; **$P < 0.01$; ***$P < 0.001$, ****$P < 0.0001$, ns = not significant. Source data are available online for this figure.

Using CRISPR/Cas9-mediated depletion or the small molecule inhibitor U73122, we confirmed prior observations (Chiang et al, 2012; Rajaiah et al, 2015; Zanoni et al, 2011) that PLCγ2 partially promoted ligand-induced TLR4 endocytosis as well as *Ifnb1* expression (Figs. 7A and EV3A–D). However, PLC activity was also required for LPS-induced *Ifnb1* expression when TLR4 endocytosis was blocked by filipin (Fig. 7A). In addition, U73122 reduced CD14 cell surface expression and abrogated LPS-induced CD14 endocytosis (Appendix Fig. S10I). Collectively, these data show that PLCγ2 controls TLR4 signaling by promoting ligand-induced TLR4 endocytosis and suggest additional roles for PLCγ2

in mediating LPS-induced CD14 endocytosis for the activation of endosomal TLR4 signaling.

Macrophage stimulation with LPS or 1Z105 increased phosphorylation of PLCγ2 at tyrosine 1217 in a manner sensitive to TAK-242 (Fig. 7B,C). As LPS-stimulation did not significantly affect total PLCγ2 expression, we concluded that TLR4 activity controls LPS-induced PLCγ2 activation. Cellular PLCγ2 protein expression increased upon 1Z105 stimulation, and 1Z105-induced increase in PLCγ2 protein was blocked by TAK-242 as well as NSC697923 (Fig. EV3E,F), indicating requirements for TLR4 activity and a possible role for ubiquitination events. We noted

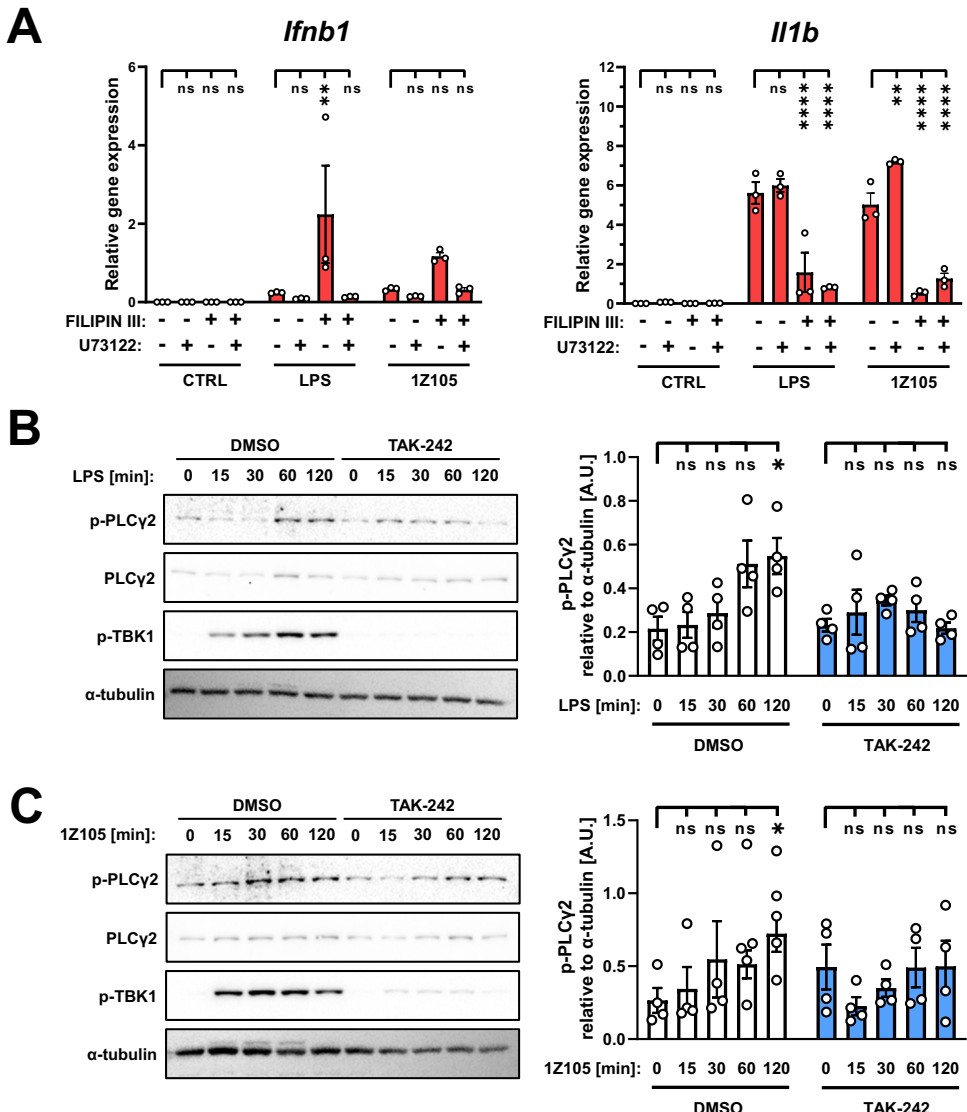

**Figure 7. PLCγ2 promotes ligand-induced TLR4 endocytosis and type I IFN expression.**

(**A**) LPS- and 1Z105-induced *Ifnb1* and *Il1b* expression (90 min) in the presence of filipin (5 μM) and/or U73122 (10 μM). (**B**, **C**) Immunoblot analysis of impact of TAK-242 (1 μM) on PLCγ2 phosphorylation at tyrosine 1217 in WT BMM upon stimulation with (**B**) LPS (100 EU/mL) or (**C**) 1Z105 (10 μM) for the indicated times. TBK1 phosphorylation was assessed as control for LPS stimulation and TAK-242 inhibitory activity. Data information: Bar plots are mean ± SEM of $n = 3$–4 biological replicates generated in independent experiments indicated as data points. Western blot images are represent 1 of $n = 4$ biological replicates performed in independent experiments. Ordinary two-way ANOVA with Dunnett's multiple comparisons test utilized in (**A**). Ordinary one-way ANOVA with Dunnett's multiple comparisons test utilized in (**B**, **C**). *$P < 0.05$; **$P < 0.01$; ***$P < 0.001$, ****$P < 0.0001$, ns = not significant. Source data are available online for this figure.

that even in resting WT macrophages treated with TAK-242 or in *Tlr4$^{-/-}$* macrophages, PLCγ2 expression was elevated (Fig. EV3G,H), suggesting that tonic TLR4 activity restricts cellular levels of PLCγ2. Collectively, these data indicate reciprocal regulatory interactions between TLR4 and PLCγ2 in macrophages.

## Discussion

TLR4 activation drives signaling pathways with distinct functional outcomes that underpin host defence, but also contribute to pathology, in infectious and inflammatory diseases (Heine and

Zamyatina, 2023). To guide the development of innovations for selective targeting of TLR4 signaling pathways, it is essential to elucidate the molecular mechanisms underlying these TLR4 signaling modalities. In this study, we made several new discoveries that revise current understanding of the mechanisms governing TLR4 endocytosis and initiation of endosomal TLR4 signaling. Specifically, TLR4-mediated type I IFN responses attributed to endosomal TLR4 activation can be uncoupled from expression and endocytosis of cell surface-expressed TLR4, showing that TLR4 endocytosis is not a prerequisite for endosomal TLR4 signaling. Moreover, TLR4 endocytosis requires (i) TLR4 activation and a functional TLR4 TIR domain, yet is independent of

canonical TLR signaling adaptor proteins, (ii) PLCγ2 activity associated with PLCγ2 phosphorylation and protein stabilisation downstream of TLR4, and (iii) activity of the ubiquitination cascade outside of canonical TLR4 signaling cascades. Based on these findings, we propose a revised model of TLR4 activation where ligand-induced TLR4 endocytosis, driven by TLR4 TIR domain activity and, in part by PLCγ2 activity, curbs pro-inflammatory TLR4-MAL-MyD88-dependent signaling at the plasma membrane and destines TLR4 for degradation (Fig. 8). TLR4 endocytosis, however, does not serve as a molecular switch to initiate endosomal TLR4 signaling. Instead, TLR4 present in intracellular compartments relies on endocytosis and directed trafficking of ligands, which in the case of LPS requires endosomal delivery through ligand-induced endocytosis of CD14 (Fig. 8).

TLR4 has been considered to occupy a unique position within the TLR family as a TLR that translocates from the cell surface to endosomes as a pre-requisite for differential signaling outputs (Heine and Zamyatina, 2023). This model was based on (i) presence of TLR4 in endosomes following LPS stimulation (Akbar et al, 2016; Husebye et al, 2006); (ii) inhibition of both LPS-induced TLR4 endocytosis and *Ifnb1* expression by the dynamin inhibitor dynasore (Kagan et al, 2008); and (iii) localization of TLR4, TRAM and TRAF3 to Rab5a-positive compartments in resting macrophages (Kagan et al, 2008). Our observations that lipid raft-disrupting agents impair pro-inflammatory gene expression and abolish TLR4 endocytosis, yet maintain *Ifnb1* expression in response to LPS, suggest that TLR4 signaling from the cell surface and endosomes arises by parallel activation of distinct pools of TLR4 at distinct subcellular locations (Fig. 8). This notion of a distinct, pre-existing intracellular pool of signaling-competent TLR4 is further supported by our data and previous findings (Takahashi et al, 2007) of LPS-induced *Ifnb1* expression in macrophages lacking cell-surface expressed TLR4. In this revised model, TLR4 endocytosis downregulates pro-inflammatory surface TLR4 signaling as originally proposed (Husebye et al, 2006), rather than as a molecular switch towards endosomal TLR4 signaling outputs (Fig. 8). TLR4 endocytosis and degradation as an "off switch" for TLR4 signaling also aligns with a reduction in seeding points for myddosomes and the overall transient nature of TLR-myddosome interactions (Fisch et al, 2024; Latty et al, 2018). This revised model is further supported by previous observations that LPS-containing liposomes activated TLR4-TRAM-TRIF-dependent signaling with minimal induction of TLR4-MAL-MyD88 signaling (Watanabe et al, 2013), and that recruitment of TLR4 from the endocytic recycling compartment to *E. coli*-containing phagosomes promoted *Ifnb1* expression (Husebye et al, 2010). Critical next steps in further defining how TLR4 functions are regulated will require elucidating how and where signaling-competent intracellular TLR4 is established and maintained. This might entail detailed exploration of TLR4 sorting and trafficking along the ER-Golgi-endosome/plasma membrane axis. Moreover, the data to date cannot exclude the possibility that in physiological contexts endocytosed TLR4 en route to degradation or TLR4 undergoing recycling may contribute to the pool of signaling-competent intracellular TLR4. Reduced LPS-induced *Ifnb1* responses in TLR4$^{N524/572Q}$- or TLR4$^{N572Q}$-expressing macrophages (Fig. 1G) might be indicative of such contributions, but could also reflect impaired per-molecule TLR4 signaling capacity due to the mutations. Intact or enhanced TLR4-driven *Ifnb1* expression when TLR4 endocytosis is abrogated

by cholesterol perturbations (Fig. 1C) would favour the latter interpretation. Elucidating the underpinning mechanisms in further detail is a worthwhile future pursuit. Nevertheless, our data establish that plasma membrane expression of TLR4 as well as TLR4 endocytosis are not a prerequisite for macrophage responses attributed to TLR4 signaling from endosomal compartments.

Originally, it was concluded that CD14 is a central regulator of both TLR4 endocytosis and TLR4-TRAM-TRIF signaling (Jiang et al, 2005; Zanoni et al, 2011). However, subsequent findings highlighted that the requirement for CD14 in TLR4-TRIF signaling and TLR4 endocytosis can be bypassed by the synthetic TLR4 agonist 1Z105 (Hayashi et al, 2014; Rajaiah et al, 2015), anti-TLR4/MD-2 agonistic monoclonal antibody UT12 (Rajaiah et al, 2015), LPS-coated latex beads (Zanoni et al, 2011) and liposome-packaged LPS (Watanabe et al, 2013). Thus, requirements for CD14 are likely attributable to its specific roles as the receptor for cellular uptake of LPS (Kitchens and Munford, 1998; Kitchens et al, 1998; Luchi and Munford, 1993), with membrane-bound, rather than soluble CD14, responsible for LPS uptake (Wright et al, 1990) and induction of type I IFN expression (Jacque et al, 2006; Saito et al, 2000; Tapping and Tobias, 1997). This notion aligns with recent findings on CD14-dependent (but TLR4-independent) trafficking of LPS into endosomes for cytosolic delivery and non-canonical inflammasome activation (Kumari et al, 2023; Vasudevan et al, 2022). These data corroborate our findings that perturbations of CD14, rather than TLR4, endocytosis mirrored effects on LPS-induced *Ifnb1* expression (Fig. 2). Collectively these observations support a model where CD14 promotes endosomal TLR4 signaling by functioning as receptor that binds LPS and traffics it to intracellular compartments containing TLR4, rather than by controlling TLR4 endocytosis and translocation to endosomal compartments (Fig. 8).

Using genetic and pharmacological approaches, our data demonstrate that TLR4 activity is essential for TLR4 endocytosis, in alignment with previous reports (Kobayashi et al, 2006; Tsukamoto et al, 2018). Nevertheless, data presented here and by others exclude a role for canonical TLR4 signaling adaptors (Rajaiah et al, 2015; Zanoni et al, 2011) and their major downstream signaling cascades in this process. Our data suggest that E1/E2 ubiquitination enzyme activity plays a role in ligand-induced TLR4 endocytosis. As canonical TLR4 signaling pathways are heavily regulated by ubiquitination (Zinngrebe et al, 2014), it would appear that the molecular events that drive TLR4 endocytosis facilitated by E1/E2 ubiquitination enzyme activity are distinct from canonical TLR4 signaling cascades. The specific ubiquitination target(s) and enzymes responsible for promotion of TLR4 endocytosis remain to be identified. Our data suggests that ubiquitination of the nine lysine residues of the TLR4 TIR domain is unlikely to be responsible for promoting TLR4 endocytosis, though we cannot exclude potential contributions by non-canonical ubiquitination (McDowell and Philpott, 2013). Based on our data and previous observations (Chiang et al, 2012; Rajaiah et al, 2015; Zanoni et al, 2011), PLCγ2 is a likely facilitator of TLR4 endocytosis downstream of TLR4 activation. PLCγ isoforms bind and, upon activation by receptor tyrosine kinases or membrane-proximal cytosolic kinases of the SYK, but also SRC (proto-oncogene tyrosine protein kinase) and TEC (tyrosine kinases expressed in hepatocellular carcinoma) families (Hajicek et al, 2019), hydrolyse phosphatidylinositol 4,5-bisphosphate (PIP$_2$) in the inner membrane leaflet, preferentially within cholesterol-rich lipid rafts (Myeong et al, 2021). Together with conversion of PIP$_2$ into phosphatidylinositol 3,4,5-

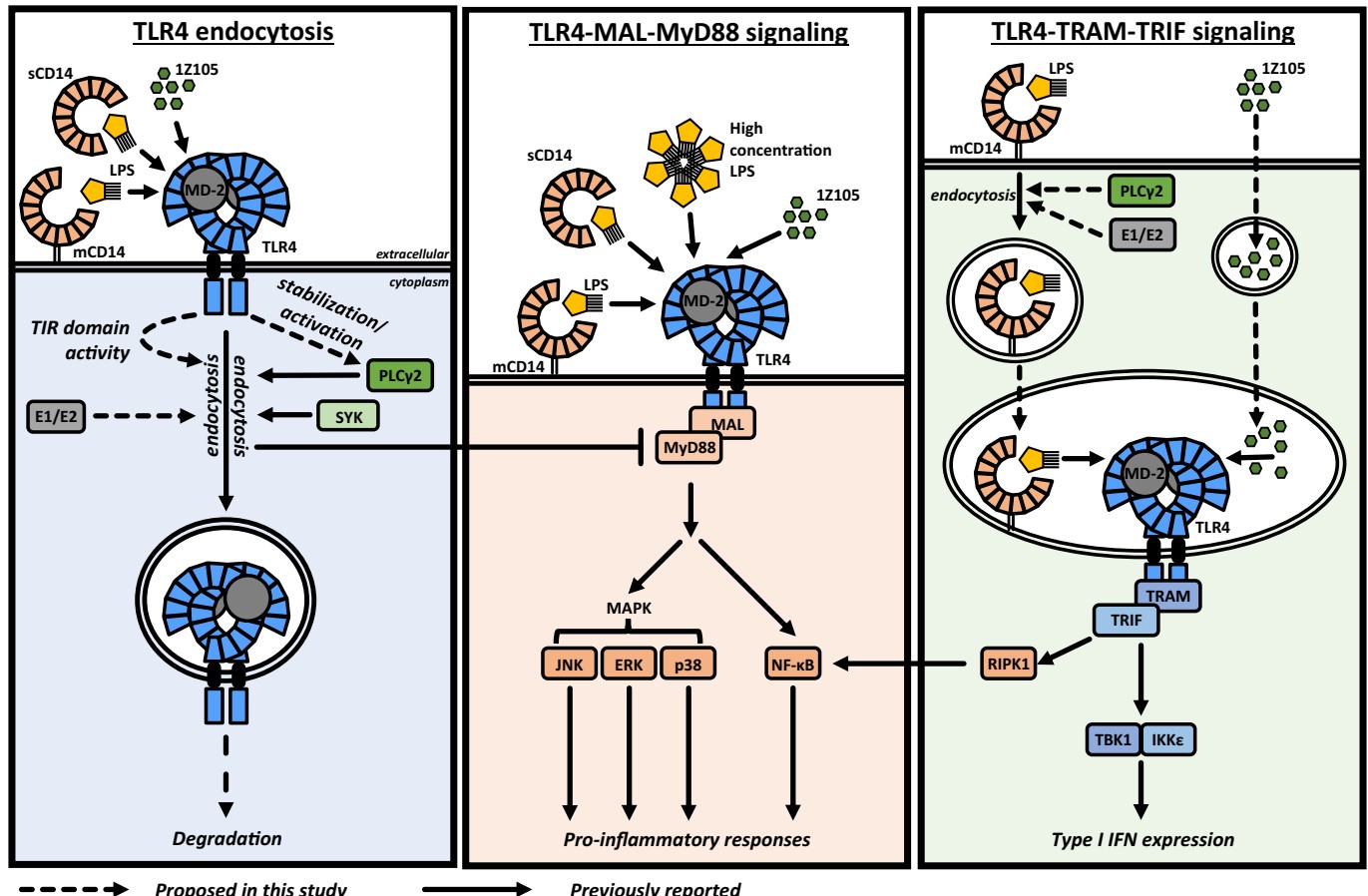

**Figure 8.    Model of TLR4 signaling modes and outcomes.**

Building on existing knowledge of TLR4 signaling, the data presented here revise current understanding of TLR4 signaling. TLR4 endocytosis is initiated upon TLR4 ligand-driven TLR4 activation. In case of the TLR4 ligand LPS, GPI-anchored membrane-bound CD14 (mCD14) (Zanoni et al, 2011), or soluble CD14 (sCD14) with lower efficiency (Tsukamoto et al, 2018), are required to transfer LPS to TLR4. In contrast, the synthetic agonist 1Z105 activates TLR4 independent of CD14 (Rajaiah et al, 2015). TLR4 endocytosis is dependent on TLR4 TIR domain activity (Figs. 3 and 4), but does not require TLR signaling adaptors MAL, MyD88, TRAM, TRIF (Rajaiah et al, 2015; Zanoni et al, 2011) (Fig. 5; Appendix Figs. S2 and S7), or BCAP (Fig. 5). Instead, E1 and E2 ubiquitination enzyme activity, independent of ubiquitination events that regulate canonical TLR4 signaling pathways, are required for TLR4 endocytosis (Fig. 6). PLCγ2 and SYK activity contributes to TLR4 endocytosis (Chiang et al, 2012; Zanoni et al, 2011) (Fig. 7; Appendix Fig. S10), and TLR4 activity regulates cellular PLCγ2 protein levels (Fig. EV3). Endocytosed TLR4 is directed towards degradation (Husebye et al, 2006) (Fig. 1). TLR4-MAL-MyD88 signaling activated at the cell surface (Latz et al, 2002) results in expression of pro-inflammatory cytokines (Kawai et al, 1999) among other cellular responses to exogenous and endogenous danger signals. CD14 is required for activation of TLR4-MAL-MyD88 signaling at low but not high LPS concentrations (Zanoni et al, 2011) with both mCD14 and sCD14 able to promote surface TLR4 activation (Frey et al, 1992; Wright et al, 1990). In contrast, the small molecule 1Z105 activates TLR4 independent of CD14 (Hayashi et al, 2014). TLR4 endocytosis negatively regulates TLR4-MAL-MyD88 signaling, thereby curbing pro-inflammatory outputs of TLR4 activation (Husebye et al, 2006; Latz et al, 2002). TLR4-MAL-MyD88 and TLR4-TRAM-TRIF signaling occur in parallel, and the latter is not reliant on TLR4 surface expression or TLR4 endocytosis (Figs. 1 and EV1). In the case of LPS, cellular uptake and trafficking requires mCD14 (Wright et al, 1990), but not sCD14 (Jacque et al, 2006; Tapping and Tobias, 1997), to deliver LPS to endosomes (Vasudevan et al, 2022). CD14 undergoes endocytosis with contributions by PLC and ubiquitination enzyme activity (Appendix Figs. S9 and S10), trafficking bound LPS into the cell (Kitchens and Munford, 1998; Kitchens et al, 1998) and transferring it to endosomal TLR4. This results in TLR4-TRAM-TRIF signaling and TBK1/IKKε activation that promotes type I IFN expression (Fig. 2). In addition, TRIF and RIPK1 promote prolonged NF-κB activation (Cusson-Hermance et al, 2005; Sato et al, 2003). As LPS is a cell-impermeable molecule (Guerville and Boudry, 2016), the requirement for mCD14 for TLR4-TRAM-TRIF signaling cannot be overridden by high LPS concentrations (Lloyd-Jones et al, 2008) or complexing with sCD14 (Saito et al, 2000). In contrast to LPS, 1Z105 does not require CD14 endocytosis to be delivered to endosomal TLR4. Nevertheless, perturbation of 1Z105-induced *Ifnb1* expression by endocytosis inhibitors (Fig. 2) and concomitant induction of LPS- and 1Z105-induced *Ifnb1* expression (Appendix Fig. S5) suggests that 1Z105 is actively trafficked to endosomal compartments where it activates TLR4-TRAM-TRIF signaling.

bisphosphate (PIP₃) by type I PI3K (Aksoy et al, 2012), this may result in local $PIP_2$ depletion. One proposed possible consequence of this depletion is diminished availability of MAL within lipid rafts for interactions with TLR4 (Kagan and Medzhitov, 2006). Such an outcome could favour TLR4 activity within lipid rafts, independent of interactions with canonical TLR signaling adaptors. A product of PLCγ-mediated $PIP_2$ hydrolysis is inositol triphosphate (IP3), which

mobilizes intracellular $Ca^{2+}$, a process initially concluded to underpin the $Ca^{2+}$-dependency of TLR4 endocytosis and endosomal TLR4 signaling (Chiang et al, 2012). More recent observations, however, implicated $Ca^{2+}$ influx via the TRPM7 channel as a facilitator of TLR4 activation (cell-surface and endosomal) upon LPS stimulation (Schappe et al, 2018). As this required CD14, but not TLR4, TRPM7 functions in TLR4 signaling and endocytosis might be specific to

CD14-dependent TLR4 agonists. Thus, the mechanisms of TLR4 activation outside of TLR adaptor engagement and placement of PLCγ2 stability and/or activation in the events that facilitate TLR4 endocytosis will be the subject of future investigation. The latter will benefit from further interrogation of the potential differential contributions of PLCγ2 to TLR4 and CD14 endocytosis, particularly in the context of putative CD14-mediated ligand trafficking. We anticipate that such insights will inform on molecular mechanisms governing subcellular localization, and related functional outcomes, of other members of the TLR family.

TLR4 has central roles in the host defence against pathogens, as well as inflammatory responses in infectious and non-infectious contexts (Heine and Zamyatina, 2023). The latter includes contributions of TLR4 to inflammation and immunopathology in diseases such as influenza (Shirey et al, 2013), rheumatoid arthritis (Abdollahi-Roodsaz et al, 2007), and autoimmune hepatitis (Hsu et al, 2017). Accordingly, TLR4 is a target of interest for therapeutic interventions (Anwar et al, 2019). Thus far, agents targeting TLR4, such as Eritoran (Opal et al, 2013) and TAK-242 (Rice et al, 2010), appear to ablate TLR4 function regardless of its subcellular localisation. Revised understanding of TLR4 signaling presented here, with discrete molecular mechanisms governing pro-inflammatory TLR4 signaling, type I IFN responses and TLR4 endocytosis (Fig. 8), might provide a rationale for nuanced, specifically targeted approaches to modulating TLR4 functions in a clinical setting. The exquisite substrate specificity of the E1/E2/E3 enzyme cascade encourages identification of the E3 ubiquitin ligase and its target(s) governing TLR4 endocytosis, as it holds potential for precision targeting of TLR4 endocytosis for therapeutic benefit.

## Limitations of this study

Flow cytometric analysis of ligand-induced decline in TLR4 surface expression using an antibody that recognizes TLR4 regardless of conformational changes in the receptor upon ligand binding (Akashi et al, 2003) is the accepted standard practice in the field (Kagan et al, 2008; Schappe and Desai, 2018). However, this approach is a measure of net surface TLR4 expression that may reflect the consequences of dynamic TLR4 trafficking to and from the cell surface. Our data indicate that endocytosed TLR4 is predominantly degraded, but whether some is recycled and or contributes to potential additional signaling activities associated with distinct intracellular compartments remains to be defined. Strategies for effectively labeling endogenous TLR4 (while maintaining full TLR4 activity) will be invaluable for tracking the fate of single TLR4 molecules across subcellular compartments and interactions with regulators of TLR4 activity. Whereas our data indicate that ubiquitination at lysine residues in the TLR4 TIR domain does not control TLR4 endocytosis, we cannot exclude potential contributions of non-canonical ubiquitination of the TLR4 TIR domain. The murine TLR4 TIR domain contains 9 serine, 8 tyrosine, 7 threonine and 3 cysteine residues. Approaches targeting these residues through single or combined mutation to assess relevance to ligand-induced TLR4 endocytosis will need to ensure that TLR4 signalling capacity is retained. In human monocytes, surface TLR4 undergoes endocytosis in response to LPS stimulation (Husebye et al, 2006). As humans are significantly more sensitive to endotoxin challenge than mice (Munford, 2010), further expansion of knowledge of TLR4 signaling modes and mechanisms governing TLR4 endocytosis in human cells will facilitate establishing relevance to human health and TLR4-driven pathologies.

# Methods

### Reagents and tools table

| Reagent/resource | Reference or source | Identifier or catalog number |
|---|---|---|
| **Experimental models** | | |
| BMM, C57BL/6N (WT) | Blumenthal lab, UQ Frazer Institute | |
| BMM, C57BL/6J (WT) | Animal Resources Centre, Perth, WA | |
| BMM, *Tlr4*⁻/⁻ | Blumenthal lab, UQ Frazer Institute | |
| BMM, *Myd88*⁻/⁻ | Blumenthal lab, UQ Frazer Institute | |
| BMM, *Cd14*⁻/⁻ | Ulett lab, Griffith University | |
| BMM, *Mal*⁻/⁻ | Ashley Mansell, while at Hudson Institute of Medical Research | |
| BMM, *Trif*⁻/⁻ | QIMR Berghofer, Brisbane | |
| BMM, *Pik3ap1*⁻/⁻ | Hamerman lab, Benaroya Research Institute | |
| BMM, C3H/HeN | Wells lab, UQ Frazer Institute | |
| BMM, C3H/HeJ | Animal Resources Centre, Perth, WA | |
| iBMM, WT | BEI Resources, NIAID, NIH | NR-9456 |
| iBMM, TLR4-knockout | BEI Resources, NIAID, NIH | NR-9458 |
| iBMM, CD14-knockout | BEI Resources, NIAID, NIH | NR-9570 |
| iBMM, MyD88-knockout | BEI Resources, NIAID, NIH | NR-15633 |
| iBMM, MAL-knockout | BEI Resources, NIAID, NIH | NR-9459 |
| iBMM, TRIF-knockout | BEI Resources, NIAID, NIH | NR-9566 |
| iBMM, TRAM-knockout | BEI Resources, NIAID, NIH | NR-9909 |
| iBMM, MyD88/TRIF knockout | BEI Resources, NIAID, NIH | NR-15632 |
| iBMM, TRIF/TRAM-knockout | BEI Resources, NIAID, NIH | NR-9568 |
| RAW264.7 cell line | ATCC | |
| Plat-E cell line | Brooks lab, UQ Frazer Institute | PMID: 10871756 |
| *Escherichia coli* MG1655 | Kindly provided by Prof Mark Schembri, The University of Queensland, Brisbane, Australia | |
| *Escherichia coli* EC958 | Kindly provided by Associate Professor Makrina Totsika while at Institute of Health and Biomedical Innovation, Queensland University of Technology, Brisbane, Australia | |

| Reagent/resource | Reference or source | Identifier or catalog number |
|---|---|---|
| *Escherichia coli* CFT073 | Kindly provided by Prof Mark Schembri, The University of Queensland, Brisbane, Australia | |
| *Listeria monocytogenes* 10403S | Kindly provided by Prof Eric Pamer while at Memorial Sloan Kettering Cancer Centre, New York, USA | |
| **Recombinant DNA** | | |
| pDONR221 | Life Technologies | |
| pEF-DEST51 | Life Technologies | |
| pMX-GW-PGK-PuroR-GFP retroviral destination vector | Brooks lab, UQ Frazer Institute | PMID: 31697803 |
| pEF-mTLR4-V5-HIS and mutant/tagged versions | This study | |
| pMX-mTLR4-V5-HIS and mutant/tagged versions | This study | |
| **Antibodies** | | |
| APC anti-mouse TLR4, clone Sa15-21 | Biolegend | #145406, RRID:AB_2562503 |
| APC Rat IgG2aκ isotype control, clone RTK2758 | Biolegend | #400512, RRID:AB_2814702 |
| Biotin anti-mouse CD14, clone Sa14-2 | Biolegend | #123306, AB_940586 |
| Biotin anti-mouse TLR4, clone Sa15-21 | Biolegend | #145409, RRID:AB_2566030 |
| Biotin Rat IgG2aκ isotype control, clone RTK2758 | Biolegend | #400504, RRID:AB_2783537 |
| streptavidin AlexaFluor 647 | Biolegend | #405237 |
| Biotin anti-mouse CD14, clone Sa2-8 | Thermo Fisher Scientific | #13-0141-82, RRID:AB_466370 |
| Mouse CD16/CD32 Fc Block, clone 2.4G2 | BD Biosciences | #553142, RRID:AB_394656 |
| α-tubulin, clone 11H10 | Cell Signaling Technologies | #2125, RRID:AB_2619646 |
| phospho-ERK1/2 | Cell Signaling Technologies | #9101, RRID:AB_331646 |
| total ERK1/2 | Cell Signaling Technologies | #9102, RRID:AB_330744 |
| phospho-JNK | Cell Signaling Technologies | #9251, RRID:AB_331659 |
| Total JNK | Cell Signaling Technologies | #9252, RRID:AB_2250373 |
| phospho-NF-κB p65 (Ser536) | Cell Signaling Technologies | #3033, RRID:AB_331284 |
| NF-κB p65 | Cell Signaling Technologies | #3034, RRID:AB_330561 |
| phospho-TBK1/NAK (Ser172) | Cell Signaling Technologies | #5483, RRID:AB_10693472 |
| TBK1/NAK | Cell Signaling Technologies | #3504, RRID:AB_2255663 |
| phospho-PLCγ2 (Tyr1217) | Cell Signaling Technologies | #3871, RRID:AB_2299548 |

| Reagent/resource | Reference or source | Identifier or catalog number |
|---|---|---|
| total PLCγ2 | Cell Signaling Technologies | #3872, RRID:AB_2299586 |
| V5-Tag, clone D3H8Q | Cell Signaling Technologies | #13202, RRID:AB_2687461 |
| anti-rabbit HRP secondary antibody | Cell Signaling Technologies | (#7074, RRID:AB_2099233 |
| total Syk, clone D3Z1E | Cell Signaling Technologies | #13198, RRID:AB_2687924 |
| total PI3K p110δ, clone EPR386 | Abcam | ab109006, RRID:AB_10859958 |
| **Oligonucleotides and other sequence-based reagents** | | |
| Site-directed mutagenesis primers | This study | Appendix Table S2 |
| qRT-PCR primers | This study | Appendix Table S3 |
| CRISPR/Cas9 crRNA guides | This study | Appendix Table S4 |
| **Chemicals, enzymes and other reagents** | | |
| TAK-242 (CLI-095) | InvivoGen | Cat# tlrl-cli95 |
| Filipin III from *Streptomyces filipinensis* | Sigma-Aldrich | Cat# F4767 |
| Methyl-β-cyclodextrin | Sigma-Aldrich | Cat# C4555 |
| Dynasore hydrate | Sigma-Aldrich | Cat# 7693 |
| EIPA | Cayman Chemicals | Cat# 14406 |
| Pitstop-2 | Sigma-Aldrich | Cat# SML1169 |
| Prochlorperazine | Sigma-Aldrich | Cat# P9178 |
| BAY 11-7082 | Selleck Chemicals | Cat# S2913 |
| IKK-16 | Selleck Chemicals | Cat# S2882 |
| MRT67307 | Selleck Chemicals | Cat# S1102 |
| U0126 | Selleck Chemicals | Cat# S1460 |
| SP600125 | Selleck Chemicals | Cat# S1076 |
| SB203580 | Selleck Chemicals | Cat# S7948 |
| PYR-41 | Selleck Chemicals | Cat# S7129 |
| NSC697923 | Selleck Chemicals | Cat# S7142 |
| MG132 | Selleck Chemicals | Cat #S2619 |
| TAK-981 | Cayman Chemicals | Cat# 32741 |
| MLN4924 | Cayman Chemicals | Cat# 15217 |
| MLN7243 | Cayman Chemicals | Cat# 30108 |
| U73122 | Selleck Chemicals | Cat# S8011 |
| CAL-101 | Selleck Chemicals | Cat# S2226 |
| R406 | Selleck Chemicals | Cat# S1533 |
| BAY 61-3606 | Selleck Chemicals | Cat# S7006 |
| Piceatannol | Tocris | Cat# 1554 |
| SC-514 | Enzo Life Sciences | Cat# BML-EI343 |
| LPS-EB Ultrapure from *E. coli* O111:B4 | InvivoGen | Cat# tlrl-3pelps |
| Pam₃CSK₄ | InvivoGen | Cat# tlrl-pms |
| poly I:C HMW | InvivoGen | Cat# tlrl-pic |

| Reagent/resource | Reference or source | Identifier or catalog number |
|---|---|---|
| ultrapure flagellin from *Pseudomonas aeruginosa* | InvivoGen | Cat# tlrl-pafla |
| CpG DNA (ODN 1668) | InvivoGen | Cat# tlrl-1668 |
| 1Z105 | Synthesized and kindly provided by Dennis Carson and Howard Cottam, University of California, San Diego | PMID: 24893985 |
| 1Y88 | Synthesized and kindly provided by Dennis Carson and Howard Cottam, University of California, San Diego | PMID: 24893985 |
| Dextran-FITC (MW 59-70 kDa) | Sigma-Aldrich | Cat# FD70S |
| murine transferrin-FITC | Jackson Immunoresearch | Cat# 015-090-050, RRID: AB_2337209 |
| SpyCatcher3 | Bio-Rad | Cat# TZC025 |
| Ampicillin | Sigma-Aldrich | Cat# A0166 |
| Kanamycin | Sigma-Aldrich | Cat# K1377 |
| Puromycin | Sigma-Aldrich | Cat# P8833 |
| Blasticidin | InvivoGen | Cat# ant-bl-1 |
| Phosphate-buffered saline (PBS) | Life Technologies | Cat# 20012050 |
| fetal bovine serum (FBS) | Life Technologies | Cat# 10099141 |
| heat-inactivated newborn calf serum (NBCS) | Life Technologies | Cat# 26010074 |
| DMEM | Life Technologies | Cat# 11960044 |
| OptiMEM | Life Technologies | Cat# 31985070 |
| HEPES | Life Technologies | Cat# 15630080 |
| sodium pyruvate | Life Technologies | Cat# 11360070 |
| L-glutamine | Life Technologies | Cat# 25030081 |
| Lipofectamine RNAiMAX | Life Technologies | Cat# 13778150 |
| Lipofectamine 2000 | Life Technologies | Cat# 11668019 |
| Millex-HP Syringe Filter Unit, 0.45 µm, polyethersulfone, 33 mm, gamma sterilized | Millipore | Cat# SLHPR33RS |
| Alt-R CRISPR-Cas9 crRNA | Integrated DNA Technologies | Custom synthesis |
| Alt-R *S.p.* Cas9 Nuclease V3 | Integrated DNA Technologies | Cat# 1081060 |
| Alt-R CRISPR-Cas9 tracrRNA | Integrated DNA Technologies | Cat# 1072532 |
| Nuclease Free Duplex Buffer | Integrated DNA Technologies | Cat# 11-01-03-01 |
| SuperSignal West Dura Extended Duration Substrate | Thermo Fisher Scientific | Cat# 34076 |
| Clarity Western ECL Substrate | Bio-Rad | Cat# 1705061 |
| PageRuler Prestained Protein Ladder, 10 to 180 kDa | Thermo Fisher Scientific | Cat# 26616 |

| Reagent/resource | Reference or source | Identifier or catalog number |
|---|---|---|
| Immobilon-P PVDF Membrane | Millipore | IPVH00010 |
| Paraformaldehyde (16% solution) | Electron Microscopy Sciences | Cat #C004 |
| iScript cDNA Synthesis kit | Bio-Rad | Cat# 1708891 |
| SYBR Green Real-Time PCR Master Mix | Applied Biosystems | Cat# 4312704 |
| RNase-free DNase I | New England Biolabs | Cat# M0303 |
| 2-propanol, for molecular biology | Sigma-Aldrich | Cat# I9516 |
| Ethanol, for molecular biology | Sigma-Aldrich | Cat# E7023 |
| Nuclease-Free Water (not DEPC-Treated) | Thermo Fisher Scientific | Cat# AM9937 |
| PureLink HiPure Plasmid Maxiprep Kit | Thermo Fisher Scientific | Cat# K210007 |
| Polybrene (Hexadimethrine bromide) | Sigma-Aldrich | Cat# H9268 |
| 5 mL Round Bottom Polystyrene Test Tube, with Snap Cap, Sterile | Corning | Cat# 352054 |
| 150 mm TC-treated Culture Dish | Corning | Cat# 430599 |
| 50 mm×15 mm standard style not Treated Bacteriological Petri dish, sterile | Corning | Cat# 351058 |
| Q5 Hot Start High-Fidelity DNA Polymerase | New England Biolabs | Cat# M0493 |
| KLD Enzyme Mix | New England Biolabs | Cat# M0554S |
| IL-6 Mouse ELISA Kit | BD | Cat# 555240 |
| Mouse IL-12p40 ELISA | BD | Cat# 555165 |
| Mouse CXCL10/IP-10/CRG-2 DuoSet ELISA | R&D Systems | Cat# DY466 |
| Mouse Interferon β, IFN-β/IFNB ELISA Kit | CUSABIO | CSB-E04945 |
| Gateway LR Clonase™ II Enzyme mix | Thermo Fisher Scientific | Cat# 11791020 |
| Gateway BP Clonase™ II Enzyme mix | Thermo Fisher Scientific | Cat# 11789020 |
| **Software** | | |
| QuantStudio Real-Time PCR Software | Applied Biosystems | |
| Adobe Illustrator | Adobe | |
| FlowJo | FlowJo, LLC | |
| GraphPad Prism | GraphPad Software | |
| ImageJ | NIH | |
| Benchling | Benchling | |
| NEBasechanger | New England Biolabs | |
| **Other** | | |
| QuantStudio 7 Flex | Applied BioSystems | |

| Reagent/resource | Reference or source | Identifier or catalog number |
|---|---|---|
| ChemiDoc Imaging System | Bio-Rad | |
| iBright CL1500 imaging system | Thermo Fisher Scientific | |
| LSR Fortessa X-20 flow cytometer | BD | |
| FACSAria Fusion cell sorter | BD | |
| GenePulser MXcell | Bio-Rad | |
| TransBlot Turbo semi-dry transfer system | Bio-Rad | |

## Study design

The molecular mechanisms underlying TLR4 endocytosis and signaling in response to stimulation with LPS, other TLR4 ligands, and bacterial infection, were investigated in murine primary macrophages, immortalized bone marrow-derived macrophages, or the macrophage-like cell line RAW264.7. The in vitro studies utilized both genetic and pharmacological approaches for interrogation of the molecular requirements of TLR4 endocytosis and TLR4-TRIF signaling. Experimental readouts included flow cytometry, qRT-PCR, immunoblotting and ELISA. At least three independent experiments were performed per dataset, unless indicated otherwise. Individual datapoints on graphs represent independent experiments. Sample sizes are detailed in figure legends. Quantification of flow cytometry data, unless specified, is presented as median fluorescence intensity ($\Delta$MFI) data with subtraction of the isotype control signal (without further normalisation or other processing).

## Mice

C57BL/6N (wild-type, WT), $Tlr4^{-/-}$, $Myd88^{-/-}$, $Cd14^{-/-}$, $Mal^{-/-}$, $Trif^{-/-}$, $Pik3ap1^{-/-}$ and C3H/HeN mice were bred under specific pathogen-free conditions. C57BL/6J (WT control for $Tlr4^{-/-}$, $MyD88^{-/-}$, $Trif^{-/-}$, $Mal^{-/-}$, $Pik3ap1^{-/-}$ and $Cd14^{-/-}$ mice) and C3H/HeJ mice were purchased from Animal Resources Centre, Perth, Australia. $Trif^{-/-}$ mice were obtained from QIMR Berghofer (Brisbane, Australia). $Mal^{-/-}$ mice were obtained from Ashley Mansell, Hudson Institute of Medical Research (Melbourne, Australia). All mice were age- and sex-matched within experiments. All animal procedures adhered to the guidelines of the National Health and Medical Research Council (NHMRC) Australian Code for the Care and Use of Animals for Scientific Purposes and were approved by The University of Queensland Animal Ethics Committee (DI/058/19, DI/059/19, DI/524/22).

## Reagents

LPS-EB Ultrapure from *E. coli* O111:B4 (Cat# tlrl-3pelps), Pam$_3$CSK$_4$ (Cat# tlrl-pms), poly I:C HMW (Cat# tlrl-pic), ultrapure flagellin from *Pseudomonas aeruginosa* (Cat# tlrl-pafla) and CpG DNA (ODN 1668) (Cat# tlrl-1668) were from InvivoGen.

1Z105 and the inactive control compound 1Y88 were synthesized and kindly provided by Dennis Carson and Howard Cottam, University of California, San Diego (Hayashi et al, 2014). Small molecule inhibitor product numbers, working concentrations and solvents are listed in Appendix Table S1: TAK-242 (CLI-095) was from InvivoGen; BAY 11-7082, IKK-16, SB203580, MRT67307, PYR-41, NSC697923, U0126, SP600125, MG132, U73122, CAL-101, R406 and BAY 61-3606 were from Selleck Chemicals; EIPA, TAK-981, MLN4924 and MLN7243 were from Cayman Chemicals; Filipin III from *Streptomyces filipinensis*, methyl-β-cyclodextrin, Pitstop-2, prochlorperazine and dynasore hydrate were from Sigma-Aldrich; Piceatannol was from Tocris; SC-514 was from Enzo Life Sciences. Dextran-FITC (MW 59–70 kDa) was from Sigma-Aldrich (#FD70S); murine transferrin-FITC was from Jackson Immunoresearch (Cat# 015-090-050, RRID: AB_2337209). SpyCatcher3 was from Bio-Rad (#TZC025). Ampicillin (Cat# A0166), kanamycin (Cat#K1377) and puromycin (Cat# P8833) were from Sigma-Aldrich; Blasticidin (Cat# ant-bl-1) was from InvivoGen. Phosphate-buffered saline (PBS), fetal bovine serum (FBS), heat-inactivated newborn calf serum (NBCS), DMEM, OptiMEM, HEPES, sodium pyruvate, L-glutamine and Lipofectamine RNAiMAX were from Life Technologies. FBS was heat-inactivated at 56 °C for 30–40 min.

APC anti-mouse TLR4 (clone Sa15-21, #145406, RRID:AB_2562503), APC Rat IgG2aκ isotype control (clone RTK2758, #400512, RRID:AB_2814702), Biotin anti-mouse CD14 (clone Sa14-2, #123306, AB_940586), Biotin anti-mouse TLR4 (clone Sa15-21, #145409, RRID:AB_2566030), Biotin Rat IgG2aκ isotype control (clone RTK2758, #400504, RRID:AB_2783537) and streptavidin AlexaFluor 647 (#405237) were from Biolegend. Biotin anti-mouse CD14 (clone Sa2-8, #13-0141-82, RRID:AB_466370), goat anti-rabbit AlexaFluor 647 (#A21246) and phalloidin-AlexaFluor 647 (#A22287) were from Thermo Fisher Scientific. Mouse CD16/CD32 Fc Block (clone 2.4G2, #553142, RRID:AB_394656) was from BD Biosciences. Antibodies against α-tubulin (clone 11H10, #2125, RRID:AB_2619646), phospho-ERK1/2 (#9101, RRID:AB_331646), total ERK1/2 (#9102, RRID:AB_330744), phospho-JNK (#9251, RRID:AB_331659), total JNK (#9252, RRID:AB_2250373), phospho-NF-κB p65 (Ser536) (#3033, RRID:AB_331284), NF-κB p65 (#3034, RRID:AB_330561), phospho-TBK1/NAK (Ser172) (#5483, RRID:AB_10693472), TBK1/NAK (#3504, RRID:AB_2255663), phospho-PLCγ2 (Tyr1217) (#3871, RRID: AB_2299548), total PLCγ2 (#3872, RRID:AB_2299586), V5-Tag (clone D3H8Q, #13202, RRID:AB_2687461), total SYK (#13198, RRID:AB_2687924) and anti-rabbit HRP secondary antibody (#7074, RRID:AB_2099233) were from Cell Signaling Technologies. Antibody against total PI3K p110δ (clone EPR386, ab109006, RRID:AB_10859958) was from Abcam.

## Cell culture

Primary bone marrow-derived macrophages (BMM) were generated through differentiation of bone marrow isolated from femurs of 8- to 16-weeks-old mice. Bone marrow was cultured in 15 cm Petri dishes in complete DMEM (10% FBS, 2 mM L-glutamine, 1 mM sodium pyruvate and 10 mM HEPES) supplemented with 20% L929 cell-conditioned medium (LCM) (macrophage differentiation medium). BMM were harvested on day 6 of

differentiation, after addition of fresh macrophage differentiation medium on days 2 and 4. After harvesting with 1 mM EDTA/PBS at 4 °C, BMMs were seeded overnight for experimentation on day 7 post-differentiation in complete DMEM with 10% LCM (macrophage maintenance medium). Immortalized BMM (iBMM) derived from WT (NR-9456), TLR4-knockout (NR-9458), CD14-knockout (NR-9570), MyD88-knockout (NR-15633), MAL-knockout (NR-9459), TRIF-knockout (NR-9566), TRAM-knockout (NR-9909), MyD88/TRIF-knockout (NR-15632) and TRIF/TRAM-knockout (NR-9568) mice were obtained through BEI Resources, NIAID, NIH. RAW264.7 macrophage-like cells were obtained from ATCC. Following confirmation of mycoplasma-negative status, iBMMs and RAW264.7 cells were cultured in a mycoplasma-free tissue culture suite using complete DMEM, and harvested via trypsinisation.

## Plasmids and cloning

Plasmids for retroviral and mammalian expression of mouse TLR4 were constructed via Gateway cloning. pENTR-mTLR4-V5-HIS entry vector was generated by BP clonase II-mediated recombination of pDONR221 with *attB1/2* site-flanked PCR product amplified from pEF6-mTLR4-V5-HIS (Curson et al, 2023) plasmid using Q5 Hot Start High-Fidelity Polymerase (NEB). Site-directed mutagenesis was performed on pENTR-mTLR4-V5-HIS and mutant derivatives, using Q5 Site-Directed Mutagenesis Kit (NEB) and following manufacturer protocol. Primers for site-directed mutagenesis (Appendix Table S2) were designed using NEBasechanger software. Compound lysine mutants were generated sequentially, in this order: K771, K774, K810, K817, K692, K727, K730, K663, K664. Expression vectors containing mTLR4-V5-HIS and mutants were generated by LR clonase II-mediated recombination of pENTR-mTLR4-V5-HIS and derivatives with pMX-GW-PGK-PuroR-GFP retroviral destination vector (Bridgford et al, 2020) or pEF-DEST51 (Life Technologies). Plasmid minipreps were performed using the Qiaprep Plasmid Spin Miniprep kit (Qiagen). Sanger sequencing for plasmid sequence confirmation was performed at the Australian Genome Research Facility. Transfection-grade plasmid preparations were generated using the PureLink HiPure Plasmid Maxiprep Kit (Thermo Fisher).

## Stimulations and infections

LPS-EB Ultrapure from *E. coli* O111:B4 and Pam₃CSK₄ were sonicated (water bath; 30 s) prior to dilution in serum-containing culture medium. 1Z105 and 1Y88 were reconstituted in DMSO and diluted in culture medium for stimulation. LPS stimulations were performed at 100 EU/mL, unless indicated otherwise for CD14 endocytosis assays (10000 EU/mL) or as part of dose titrations. 1Z105 stimulations were performed at 10 µM, unless indicated otherwise as part of dose titrations. Unless otherwise stated, cells were treated with small molecule inhibitors for 1 h prior to stimulation. Small molecule inhibitor working concentrations are listed in Appendix Table S1, unless indicated in dose titrations. Bacteria for infection were prepared from a 5 mL overnight 37 °C shaking culture in LB (*Escherichia coli;* MG1655 [kindly provided by Prof Mark Schembri, School of Chemistry and Molecular Biosciences, University of Queensland, Brisbane, Australia], EC958 [kindly provided by Associate Professor Makrina Totsika, Institute

of Health and Biomedical Innovation, Queensland University of Technology, Brisbane, Australia], CFT073) or BHI (*Listeria monocytogenes;* 10403S [kindly provided by Eric Pamer, Memorial Sloan Kettering Cancer Centre, New York, USA]) medium. Cultures were diluted 1:100 in 10 mL of the appropriate medium and grown to early-mid log phase ($OD_{600}$ 0.3–0.8 for *E. coli*, $OD_{600}$ 0.05–0.1 for *L. monocytogenes*). Cultures were pelleted via centrifugation ($3270 \times g$, 4 °C, 10 min), washed in PBS, pelleted again, and then resuspended in cell culture medium. Macrophages were infected at a multiplicity of infection (MOI) of 10. Bacterial inoculum was verified by plating serial dilutions on LB or BHI agar and calculating colony-forming units.

## Retrovirus production

Plat-E retroviral packaging cells were maintained in complete DMEM supplemented with 10 µg/mL Blasticidin and 1 µg/mL puromycin and harvested via trypsinisation (Life Technologies). For retrovirus production, $2.5 \times 10^6$ Plat-E cells were seeded in 10 cm tissue culture dishes in 10 mL complete DMEM without antibiotics. After overnight incubation, cells were transfected with retroviral expression plasmid (24 µg empty vector, equimolar expression vector per transfection) using Lipofectamine 2000. Medium was changed 24 h post-transfection, and retrovirus-containing supernatant was harvested 48 h post-medium change. 10 mL retroviral supernatant was supplemented with 20 mM HEPES, 5 µg/mL polybrene and 20% LCM, then passed through a 0.45 µm PES syringe filter (Millipore) to remove cell debris. BMDM on day 2 of differentiation were transduced via spinfection at $1000 \times g$, 35 °C, 2 h in non-treated six-well culture plates, followed by addition of an equal volume of complete DMEM with 20% LCM. Culture medium was replaced 48 h post-transduction, with 10 µg/mL puromycin added 72 h post-transduction. Transduced BMMs were harvested 24 h post-puromycin addition and seeded overnight without puromycin for experimentation on the next day.

## CRISPR/Cas9-mediated knock-out/-down

crRNA guides for targeting *Tlr4, Syk, Plcg2, Pik3cd* were designed using Benchling (Appendix Table S4). sgRNA was generated by combining 10 µM crRNA, 10 µM tracrRNA and nuclease-free duplex buffer (Integrated DNA Technologies) (ratio 3:3:4) in a sterile 1.5 mL microcentrifuge tube, heating at 95 °C for 5 min, then allowing to cool at room temperature for at least 20 min. Ribonucleoprotein (RNP) stock was prepared by combining 10 µM sgRNA stock, 3 µM Alt-R *S.p.* Cas9 Nuclease V3 (Integrated DNA Technologies) and OptiMEM (ratio 1:1:23), and incubating at room temperature for 5 min. Transfection mix was prepared by combining 12.5 µL RNP stock, 1.2 µL Lipofectamine RNAiMAX and 11.3 µL OptiMEM, and incubating at room temperature for 20 min. For deletion of *Tlr4*, 25 µL transfection mix was added to the bottom of a flat-bottom 96-well plate, then 125 µL RAW264.7 cell suspension ($0.16 \times 10^6$/mL) added. Transfected cells were incubated at 37 °C, 5% $CO_2$ for 48 h, then harvested and expanded for fluorescence-activated cell sorting of polyclonal TLR4-positive and -negative ($RAW^{TLR4ko}$) populations.

For CRISPR/Cas9-mediated knockdown of *Syk, Plcg2, Pik3cd* in primary macrophages, the transfection protocol was scaled up proportionally for $2 \times 10^6$ primary bone marrow-derived macrophages (2 mL at $10^6$/mL) per well in six-well plates. Cells were

incubated with transfection mix for 6 h at 37 °C, 5% $CO_2$, followed by replacement with 2 mL fresh macrophage maintenance medium. Functional studies were conducted at 72 h post-transfection. Protein knockdown was confirmed by immunoblot.

## Transfection of TLR4-deficient RAW264.7 cells

Stable transfection of TLR4-deficient RAW264.7 cells (RAW^TLR4ko) with mutant TLR4 constructs was performed via electroporation. In total, $5 \times 10^6$ cells were electroporated with 20 µg transfection-grade pEF-mTLR4-V5 plasmid using a Bio-Rad GenePulser MXcell (250 V, 1000 µF capacitance, ∞ resistance), then incubated for 48 h in fresh complete DMEM. Antibiotic selection for transfected cells was performed by adding 10 µg/mL blasticidin (InvivoGen) at 48 h post-transfection. For RAW^TLR4ko cells stably transfected with mTLR4^WT, mTLR4^ΔTIR, mTLR4^9KΔR and SpyTag003-mTLR4-V5, TLR4-positive populations were isolated using fluorescence-activated cell sorting on BD FACSAria™ Fusion cell sorter. RAW^TLR4ko cells stably transfected with mTLR4^N524/572Q and mTLR4^N572Q were analysed without further sorting, with unsorted mTLR4^WT-expressing cells used as control.

## SpyTag/SpyCatcher assays

$0.5 \times 10^6$ RAW^TLR4ko cells stably transfected with SpyTag003-mTLR4-V5 were seeded in complete DMEM in a 24-well plate and incubated overnight. Following inhibitor pre-treatment for 1 h (DMSO/TAK-242 (1 µM)), culture medium was replaced with cold complete DMEM containing 5 µM recombinant SpyCatcher003 (Bio-Rad), and incubated at 4 °C for 5 min. SpyTag003-containing medium was removed and cells washed twice with cold PBS. Following PBS washes, warm complete DMEM containing DMSO/TAK-242 (1 µM) and DMEM/LPS (100 EU/mL) was added to cells, with protein lysates collected at the indicated timepoints for subsequent immunoblot analysis.

## TLR4 and CD14 endocytosis assays

$0.3 \times 10^6$ BMM or iBMM were seeded in macrophage maintenance medium in sterile 5 mL polystyrene FACS tubes (Falcon) and incubated overnight. Following inhibitor pre-treatment and subsequent stimulation (250 µL total volume), tubes were placed on ice and 3 mL ice-cold staining buffer (3% NBCS/PBS) added to stop cell stimulation. All subsequent wash steps were conducted via centrifugation at $300 \times g$, 5 min, 4 °C using 2 mL staining buffer. Following 2 initial wash steps, cells were incubated for 1 h on ice with Fc Block (1:250) and primary antibody (TLR4-APC, 1:50; CD14-biotin, 1:500). Cells were washed twice, then fixed for 15 min in 4% paraformaldehyde/PBS. For CD14 endocytosis assays, additional staining with secondary reagent (streptavidin-Alexa-Fluor 647, 1:5000) was also conducted prior to fixation. Following fixation, cells were washed twice more, and resuspended in 200 µL staining buffer for analysis on an LSR Fortessa X-20 flow cytometer (BD). Data analysis was conducted using FlowJo (BD).

## Dextran-FITC/transferrin-FITC uptake assay

$0.3 \times 10^6$ BMM were seeded in macrophage maintenance medium in sterile 5 mL polystyrene FACS tubes and incubated overnight. Following 60 min inhibitor pre-treatment (Appendix Table S1),

cells underwent 120 min incubation with FITC-labeled cargo (Dextran-FITC at 0.5 mg/mL, Transferrin-FITC at 12.5 µg/mL; total assay volume 250 µL). Tubes were placed on ice and 3 mL ice-cold staining buffer (3% NBCS/PBS) added to stop cargo uptake. All subsequent wash steps were conducted via centrifugation at $300 \times g$, 5 min, 4 °C using 2 mL staining buffer. Following 2 initial wash steps, 1 mL staining buffer, adjusted to pH 3, was added to quench fluorescence of surface-bound FITC-cargo. Following FITC quenching, cells were washed twice more, and resuspended in 200 µL staining buffer for analysis on an LSR Fortessa X-20 flow cytometer (BD). Data analysis was conducted using FlowJo (BD).

## SDS-PAGE and immunoblotting

Samples were generated via washing of treated cells with cold PBS on ice, followed by lysis in Laemmli sample buffer containing 100 mM DTT and snap-freezing in dry ice. Prior to SDS-PAGE, samples were heated at 98 °C for 5 min, sonicated (water bath, 30 s), and centrifuged at $20,000 \times g$ for 5 min at room temperature. Protein separation was performed using 5% stacking/10% separation SDS-PAGE, followed by semi-dry (Trans-Blot, Bio-Rad) or full wet (PLCγ2 blots only) transfer onto methanol-activated PVDF membrane. Protein transfer was visualized using Ponceau S staining. After removal of Ponceau S, membranes were blocked (5% milk/TBST—most proteins; 5% BSA/TBST for detection of phospho-JNK and phospho-PLCγ2) for 1 h at room temperature, followed by 3 × 10 min washing in TBST. Primary antibody incubation was performed overnight at 4 °C following the manufacturer's recommendations for antibody concentration and diluent. Following 3 × 10 min washing in TBST, secondary antibody incubation was performed for 1 h at room temperature following the manufacturer's recommendations for antibody concentration and diluent. Membranes were developed using SuperSignal West Dura Extended Duration Substrate (Thermo Fisher Scientific) or Clarity Western ECL Substrate (Bio-Rad) and imaged using the ChemiDoc Imaging System (Bio-Rad) or iBright CL1500 imaging system (Thermo Fisher Scientific). Densitometry analysis was performed using ImageJ (NIH).

## RNA extraction and qRT-PCR

All primers were designed using Integrated DNA Technologies Real-Time PCR tool (Appendix Table S3). Cells for mRNA expression analysis ($0.5–1 \times 10^6$ in 1 ml culture medium per well, 12-well plates) were lysed in 500 µL TRIzol (Thermo Fisher), snap-frozen on dry ice, and stored at −80 °C. RNA isolation was conducted according to the manufacturer's protocol. RNA samples to be analyzed for *Ifnb1* expression were treated with RNase-free DNase I (NEB) according to manufacturer's protocol. 1 µg RNA was used to generate cDNA using the iScript cDNA Synthesis kit (Bio-Rad) in a 20 µL cDNA reaction, diluted 1:3 in nuclease-free water and used for qRT-PCR (SYBR Green Real-Time PCR Master Mix; Applied Biosystems) on a QuantStudio 7 Flex machine. Data was analyzed using QuantStudio Real-Time PCR Software (Applied Biosystems). *Hprt* expression was used as housekeeping gene for normalization of gene expression ($\Delta C_t$ method).

## Enzyme-linked immunosorbent assay (ELISA)

Cell culture supernatants for ELISA were collected, aliquoted and frozen at -80 °C. Cytokine concentrations were analysed using ELISA according to manufacturers' protocols: IL-6, IL-12p40 (BD Biosciences), IFN-β (CUSABIO), CXCL10 (DuoSet R&D Systems).

## Data processing, figure compilation and statistical analysis

Experimental data was processed using Microsoft Excel, GraphPad Prism and ImageJ. Figures were exported from their native program into PDF, EMF or SVG format, and compiled in Adobe Illustrator. Statistical analysis was performed using GraphPad Prism. One- or two-way ANOVA with appropriate multiple comparisons test, or unpaired two-tailed $t$ test, were used as indicated in figure legends. $*P < 0.05$; $**P < 0.01$; $***P < 0.001$, $****P < 0.0001$, ns = not significant. Data points on graphs represent independent experiments. Radar charts were generated using Microsoft Excel, with values for each parameter expressed as percentage induction of response relative to vehicle control. For radar chart overlap calculations, percentage of overlap between two parameters was determined by dividing the sum of the minimum value between each parameter per inhibitor treatment by the total sum of each read-out, then multiplying by 100 to obtain a percentage value. Percentage represents overlap of the parameter with the larger polygon area over the parameter with the smaller polygon area.

## Data availability

No primary datasets have been generated and deposited.

The source data of this paper are collected in the following database record: biostudies:S-SCDT-10_1038-S44319-025-00444-2.

## Peer review information

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

## Acknowledgements

The authors are grateful to Meg Donovan and other members of the Blumenthal lab for insightful discussions and critical feedback on the manuscript. We thank Dennis Carson, Howard Cottam (UC San Diego), Mark Schembri (The University of Queensland), Makrina Totsika (Queensland University of Technology), Erik Pamer (University of Chicago), and Ashley Mansell (Hudson Institute of Medical Research) for providing cells and reagents, and the animal facility at the Queensland Institute for Medical Research – Berghofer for *Trif*$^{-/-}$ mice. Immortalized BMM (iBMM) were obtained through BEI Resources, NIAID, NIH. This research was carried out at the Translational Research Institute (TRI), Woolloongabba, QLD, Australia. TRI is supported by a grant from the Australian Government. The authors thank the TRI Flow Cytometry, Microscopy and Biological Resources (BRF) core facilities, and mycoplasma detection service, for enabling this work to be performed. This study was supported by the National Health and Medical Research Council (Ideas Grant GNT1182226 to AB; Investigator Grant GNT2025931 to BK; Ideas Grant GNT2003688 to KJS) and the Australian Research Council (ARC Future Fellowship FT220100487 to AB; ARC Laureate Fellowship FL180100109 to BK).

## Author contributions

**Thomas E Schultz**: Conceptualization; Data curation; Formal analysis; Investigation; Methodology; Writing—original draft; Writing—review and editing. **Carmen D Mathmann**: Formal analysis; Investigation; Methodology; Writing—review and editing. **Leslie C Domínguez Cadena**: Investigation. **Timothy W Muusse**: Investigation. **Hyoyoung Kim**: Resources. **James W Wells**: Resources; Writing—review and editing. **Glen C Ulett**: Resources. **Jessica A Hamerman**: Resources. **Andrew J Brooks**: Resources; Writing—review and editing. **Bostjan Kobe**: Resources; Writing—review and editing. **Matthew J Sweet**: Resources; Writing—review and editing. **Katryn J Stacey**: Resources; Supervision; Methodology; Writing—review and editing. **Antje Blumenthal**: Conceptualization; Resources; Supervision; Visualization; Methodology; Writing—original draft; Project administration; Writing—review and editing.

Source data underlying figure panels in this paper may have individual authorship assigned. Where available, figure panel/source data authorship is listed in the following database record: biostudies:S-SCDT-10_1038-S44319-025-00444-2.

## Disclosure and competing interests statement

The authors declare no competing interests.

# Expanded View Figures

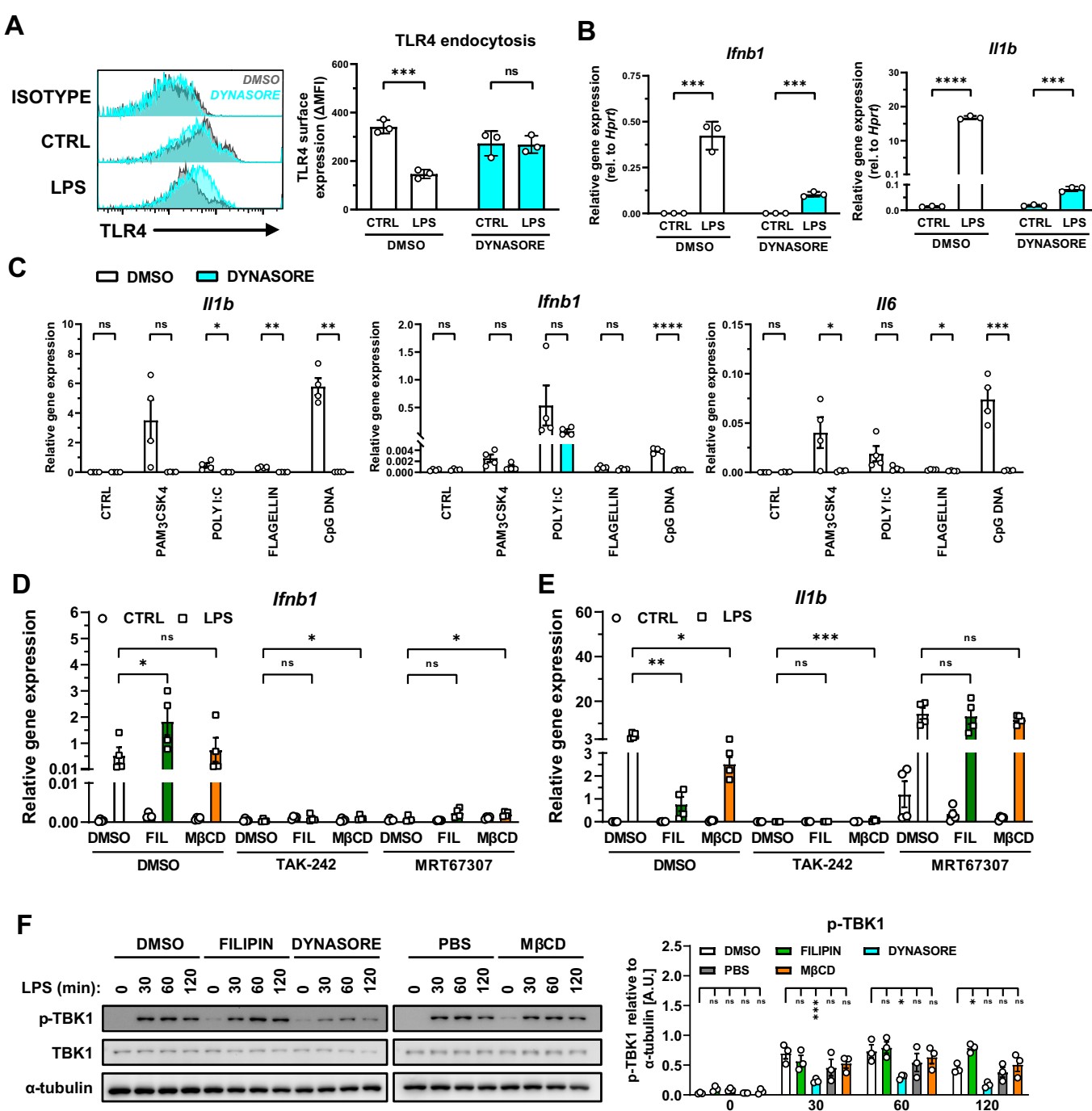

**Figure EV1. Impact of dynasore and lipid raft inhibitors on TLR-mediated macrophage activation.**

(A) Flow cytometric analysis of TLR4 surface expression in WT BMM treated for 60 min with DMSO or dynasore (80 µM), followed by stimulation with LPS (100 EU/mL) or left unstimulated (CTRL) for 120 min. (B) qRT-PCR analysis of *Ifnb1* and *Il1b* expression in WT BMM treated for 60 min with DMSO or dynasore (80 µM), followed by stimulation with LPS (100 EU/mL) for 90 min. (C) qRT-PCR analysis of *Il1b*, *Ifnb1* and *Il6* mRNA expression in WT BMM treated for 60 min with DMSO or dynasore (80 µM), followed by stimulation with Pam$_3$CSK$_4$ (10 ng/mL), poly I:C (10 µg/mL), flagellin (0.5 µg/mL) or CpG DNA (1 µM), or left unstimulated (CTRL), for 90 min. (D, E) Impact of TAK-242 (1 µM) or MRT67307 (5 µM) on LPS-induced (D) *Ifnb1* and (E) *Il1b* expression (90 min) in WT BMM treated with filipin (5 µM), MβCD (10 mM), or DMSO as control. (F) Time-course of LPS-induced TBK1 phosphorylation in WT BMM treated with filipin, MβCD, dynasore or DMSO as solvent control. Data information: Flow cytometry histograms and immunoblot images depict 1 representative of $n = 3$ biological replicates generated in independent experiments. Bar plots are mean ± SEM of $n = 3$–4 biological replicates generated in independent experiments indicated as data points. Unpaired two-tailed $t$ test performed in (A–C); ordinary two-way ANOVA with Dunnett's multiple comparison test utilized in (D–F); *$P < 0.05$; **$P < 0.01$; ***$P < 0.001$, ****$P < 0.0001$, ns = not significant. Source data are available online for this figure.

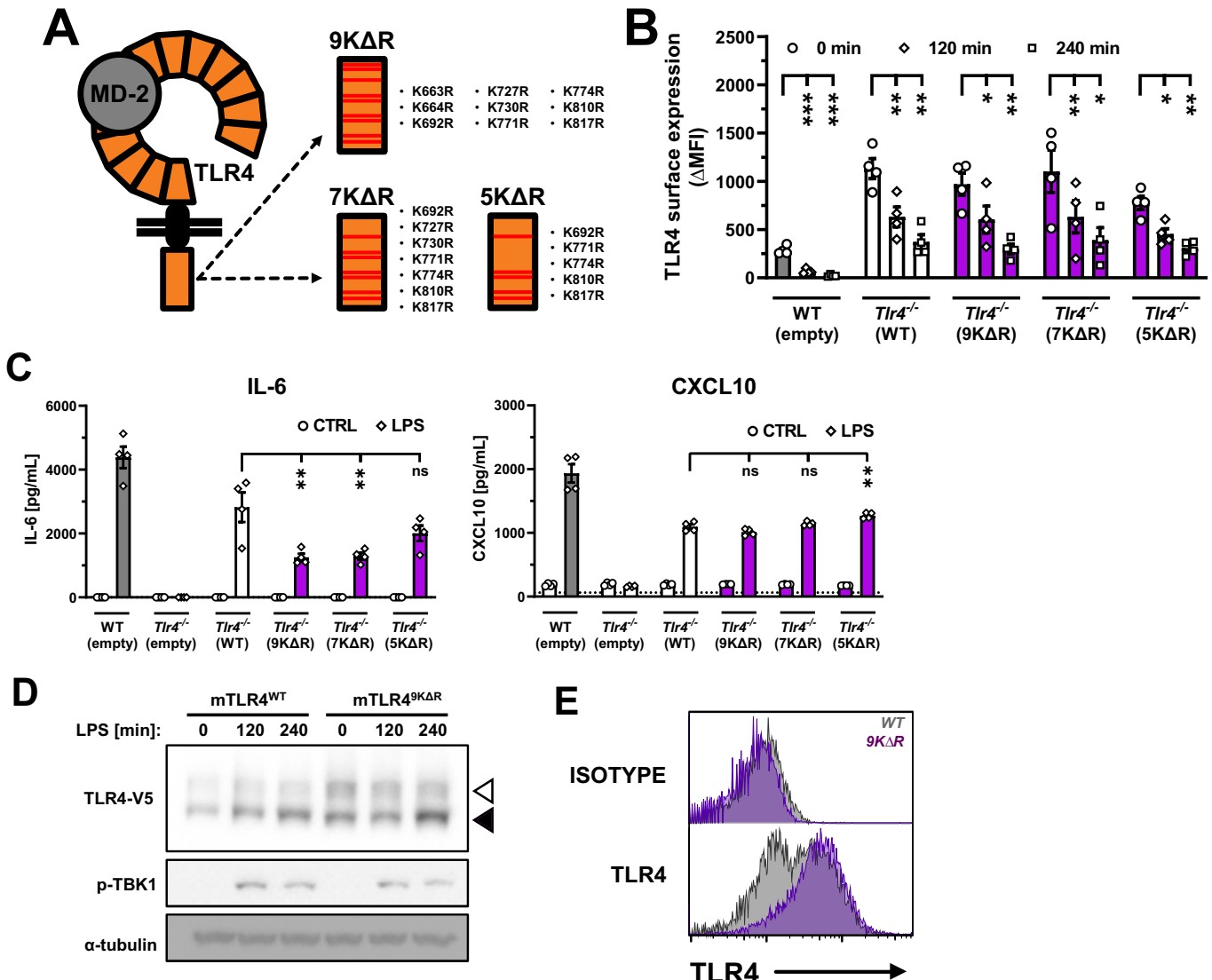

**Figure EV2.  Lysine residues in the TLR4 TIR domain were not required for LPS-induced TLR4 endocytosis.**

(A–C) *Tlr4-/-* BMMs retrovirally reconstituted with (A) mTLR4WT, mTLR49KΔR, mTLR47KΔR, mTLR45KΔR or empty vector were stimulated with LPS (100 EU/mL); (B) TLR4 surface expression assessed at 120 and 240 min post-stimulation and (C) IL-6 and CXCL10 production assessed at 24 h post-stimulation. (D, E) RAWTLR4ko cells were stably transfected with V5-tagged constructs of mTLR4WT or mTLR49KΔR. (D) Total cellular expression of TLR4-V5 during LPS stimulation (100 EU/mL, 0, 120, 1240 min) was assessed via immunoblot. Phospho-TBK1 served as control for cellular activation; a-tubulin served as loading control. Black arrow head indicates low glycosylated intracellular TLR4, white arrow head indicates highly-glycosylated cell surface-expressed TLR4. (E) TLR4 surface expression was assessed in resting cells via flow cytometry. Data information: Immunoblot images depict 1 representative of *n* = 3 biological replicates generated in independent experiments. Flow cytometry histogram represents cells post sorting. Bar plots are mean ± SEM of *n* = 3–4 biological replicates generated in independent experiments indicated as data points. RM two-way ANOVA with Dunnett's multiple comparisons test utilized in (B). Ordinary one-way ANOVA with Dunnett's multiple comparisons test utilized in (C). *P < 0.05; **P < 0.01; ***P < 0.001, ****P < 0.0001, ns = not significant. Source data are available online for this figure.

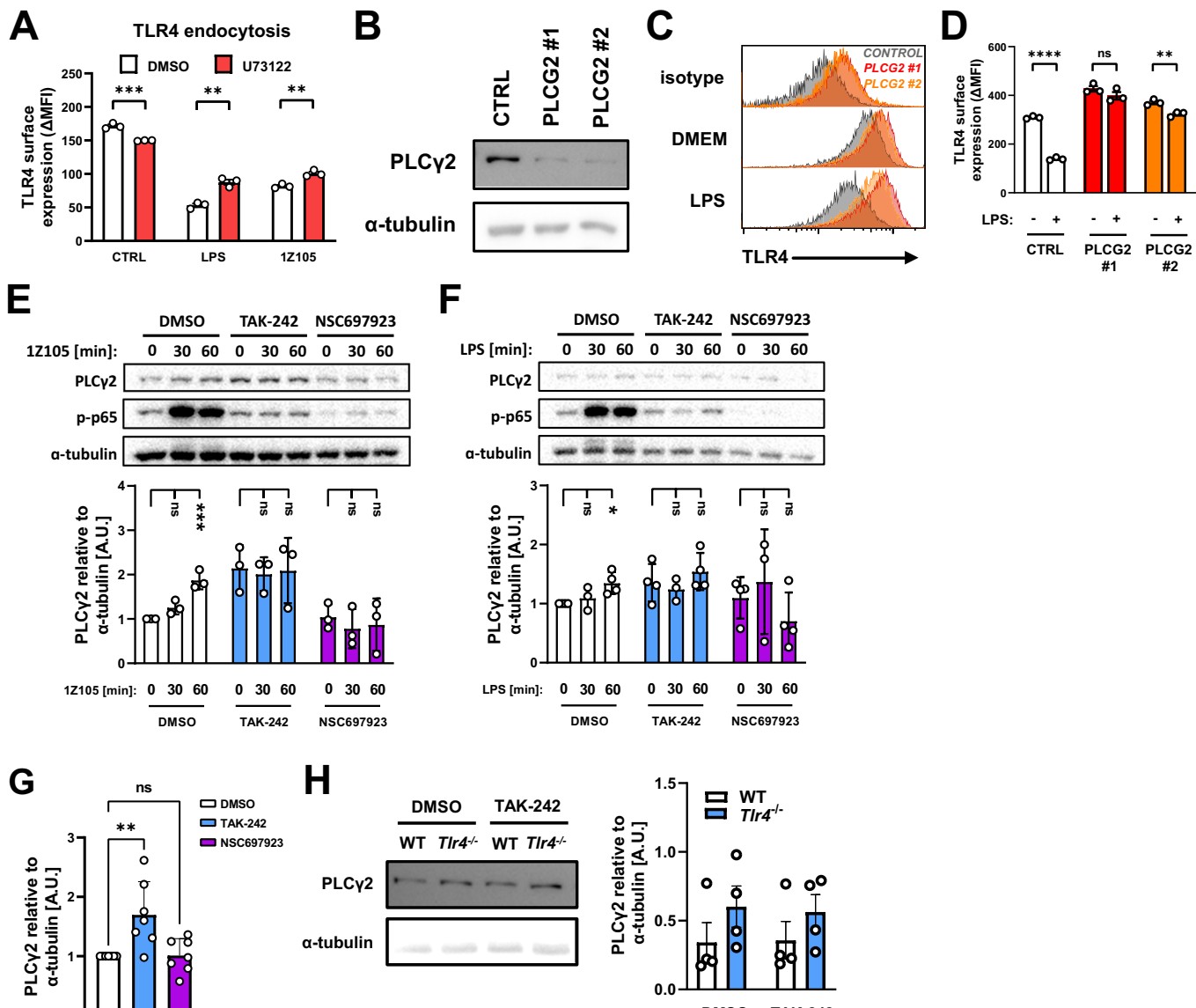

**Figure EV3. PLCγ2 regulates TLR4 endocytosis.**

(A) Impact of PLC inhibitor (U73122, 10 μM) on LPS- (100 EU/mL) and 1Z105-induced (10 μM) TLR4 endocytosis (120 min) in WT BMM. (B–D) CRISPR/Cas9-mediated knockdown of PLCγ2 expression in WT BMM (B) was verified by immunoblot analysis, and (C, D) shown to impair LPS-induced (100 EU/mL, 120 min) TLR4 endocytosis as assessed by flow cytometry. (E, F) Immunoblot analysis of the impact of TAK-242 (1 μM) or NSC694923 (10 μM) on total PLCγ2, phosphorylated NF-κB p65 and α-tubulin in WT BMM upon stimulation with (C) 1Z105 (10 μM) or (D) LPS (100 EU/mL) for the indicated times. (G) Comparison of PLCγ2 expression in unstimulated cells presented in (E, F). (H) Immunoblot analysis of PLCγ2 cellular expression in WT and Tlr4−/− BMM treated with DMSO or TAK-242 (1 μM) for 120 min. Data information: Flow cytometry histograms and immunoblot images depict 1 representative of n = 3–4 biological replicates generated in independent experiments. Bar plots are mean ± SEM of n = 3–7 biological replicates generated in independent experiments indicated as data points. Unpaired two-tailed *t* test utilized in (A, D). Ordinary one-way ANOVA with Dunnett's multiple comparisons test utilized in (E–G). *P < 0.05; **P < 0.01; ***P < 0.001, ****P < 0.0001, ns = not significant. Source data are available online for this figure.

