## [Peer Review File · EMBO Reports]

TLR4 endocytosis and endosomal signaling are distinct and independent outcomes of TLR4 activation

Thomas Schultz, Carmen Mathmann, Leslie Dominguez Cadena, Timothy Muusse, Hyoyoung Kim, James Wells, Glen Ulett, Jessica Hamerman, Andrew Brooks, Bostjan Kobe, Matthew Sweet, Katryn Stacey, and Antje Blumenthal

Corresponding author(s): Antje Blumenthal (a.blumenthal@uq.edu.au)

Review Timeline:

Submission Date:	10th Aug 24
Editorial Decision:	18th Sep 24
Revision Received:	5th Mar 25
Editorial Decision:	21st Mar 25
Revision Received:	27th Mar 25
Accepted:	28th Mar 25

Editor: Achim Breiling

Transaction Report:

Dear Dr. Blumenthal,

Thank you for the submission of your manuscript to EMBO reports. I have now received the reports from the three referees that were asked to evaluate your study, which can be found at the end of this email.

As you will see, the referees find the study very interesting. Nevertheless, they have several comments, concerns, and suggestions, indicating that a major revision of the manuscript is necessary to allow publication of the study in EMBO reports. As the reports are below, and all the concerns need to be addressed, I will not detail them further here.

Given the constructive referee comments, I would like to invite you to revise your manuscript with the understanding that the concerns of the referees must be addressed in the revised manuscript and in a detailed point-by-point response. Acceptance of your manuscript will depend on a positive outcome of a second round of review. It is EMBO reports policy to allow a single round of revision only and acceptance of the manuscript will therefore depend on the completeness of your responses included in the next, final version of the manuscript.

- 1) a .docx formatted version of the final manuscript text (including legends for main figures, EV figures and tables), but without the figures included. Figure legends should be compiled at the end of the manuscript text.
- 2) individual production quality figure files as .eps, .tif, .jpg (one file per figure), of main figures and EV figures. Please upload these as separate, individual files upon re-submission.

- 4) a complete author checklist, which you can download from our author guidelines (<https://www.embopress.org/page/journal/14693178/authorguide>). Please insert page numbers in the checklist to indicate where the requested information can be found in the manuscript. The completed author checklist will also be part of the RPF.

- 5) that primary datasets produced in this study (e.g. RNA-seq, ChIP-seq, structural and array data) are deposited in an

appropriate public database. If no primary datasets have been deposited, please also state this in a dedicated section (e.g. 'No primary datasets have been generated and deposited'), see below.

The accession numbers and database should be listed in a formal "Data Availability" section (placed after Materials & Methods) that follows the model below. This is now mandatory (like the COI statement). Please note that the Data Availability Section is restricted to new primary data that are part of this study. This section is mandatory. As indicated above, if no primary datasets have been deposited, please state this in this section

Data availability

8) Regarding data quantification and statistics, please make sure that the number "n" for how many independent experiments were performed, their nature (biological versus technical replicates), the bars and error bars (e.g. SEM, SD) and the test used to calculate p-values is indicated in the respective figure legends (also for EV figures and all those in an Appendix). Please also check that all the p-values are explained in the legend, and that these fit to those shown in the figure. Please provide statistical testing where applicable. Please avoid the phrase 'independent experiment', but clearly state if these were biological or technical replicates. Please also indicate (e.g. with n.s.) if testing was performed, but the differences are not significant. In case n=2, please show the data as separate datapoints without error bars and statistics. See also: <http://www.embopress.org/page/journal/14693178/authorguide#statisticalanalysis>

9) Please add scale bars of similar style and thickness to microscopic images, using clearly visible black or white bars (depending on the background). Please place these in the lower right corner of the images themselves. Please do not write on or near the bars in the image but define the size in the respective figure legend.

10) Please also note our reference format:

12) We now use CRedit to specify the contributions of each author in the journal submission system. CRedit replaces the author contribution section. Please use the free text box to provide more detailed descriptions and do NOT provide your final manuscript text file with an author contributions section. See also our guide to authors: <https://www.embopress.org/page/journal/14693178/authorguide#authorshipguidelines>

13) All Materials and Methods need to be described in the main text using our 'Structured Methods' format, which is required for

all research articles. According to this format, the Materials and Methods section should include a Reagents and Tools Table (listing key reagents, experimental models, software, and relevant equipment and including their sources and relevant identifiers), uploaded as separate file, followed by a Methods and Protocols section in which we encourage the authors to describe their methods using a step-by-step protocol format with bullet points, to facilitate the adoption of the methodologies across labs. More information on how to adhere to this format as well as downloadable templates (.doc) for the Reagents and Tools Table can be found in our author guidelines (section 'Structured Methods'):

14) Please add up to 5 keywords to the manuscript and order the manuscript sections like this, using these names: Title page - Abstract - Keywords - Introduction - Results - Discussion - Methods - Data availability section - Acknowledgements (including funding information) - Disclosure and Competing Interests Statement - References - Figure legends - Expanded View Figure legends

I look forward to seeing a revised form of your manuscript when it is ready.

Yours sincerely,

Referee #1:

The study by Schultz and colleagues explores in depth the link between endosomal TLR signalling and TLR4 endocytosis. Previously published work had led to the acceptance of a model of TLR4 signalling whereby endocytosis of TLR4 was required for endosomal TLR4 signalling. The data presented in this study challenges this model and reveals instead that endocytosis of TLR4 is not required for endosomal TLR4 signalling. Endosomal TLR4 signalling is associated with IRF activation and the antiviral response while cell surface TLR4 signalling is associated with pro-inflammatory cytokine expression.

Understanding the molecular mechanisms distinguishing these two outputs of TLR4 activation are important for our understanding of innate immune responses and how they could be manipulated. The authors present data showing that inhibitors of endocytosis previously show off-target effects that undermine the conclusions of previous studies on TLR endocytosis. The authors also show that CD14 endocytosis is required for the transfer of the TLR4 ligand LPS to endosomal TLR4. The authors show that TLR4 activation is required for endocytosis, but that this occurs independently of TLR adaptors and the MAPK and TBK1 pathways activated by TLR4. Instead the authors propose that ubiquitination as well as PLC γ 2 is required for endocytosis of TLR4.

Overall the data is of high quality and the conclusions are mostly justified. The approach is thorough and incorporates both pharmacological inhibitors and genetic knockout cells. If anything the figures are possibly too dense and at times difficult to interpret due to the volume of data presented, particularly when the supplementary figures are included. The authors should see if there is any non-essential data that could be removed to improve the readability of the study.

Major points:

1. To evaluate endocytosis the authors use a decrease in cell surface TLR4 expression as measured by flow cytometry immunofluorescence analysis. Can the authors rule out changes in total TLR4 protein expression that may contribute to decreased cell surface staining in any of their experiments?
2. The authors state the E1 and E2 enzyme inhibitors blocked LPS-induced CD14 endocytosis without impacting other cellular endocytic uptake mechanisms - however the data shown in Figure S9D shows that the E2 inhibitor NSC697923 inhibited

Dextran FITC uptake while NSC697923 and the E2 inhibitor BAY 11-7082 both inhibited Transferrin uptake. Furthermore, the Figure S9 does not contain any analysis of the effect of the E1 inhibitor PYR-41 in the same assays. Thus, the authors conclusions are not supported by the data and should be modified to accurately reflect the data.

3. The authors propose that TLR4 TIR domain activity controls the cellular levels of PLC γ 2 which is also needed for TLR4 endocytosis. Do TLR4^{-/-} cells have reduced levels of PLC γ 2? which would be expected if this were the case.

4. While the data presented in Figure 7 show statistically significant increased PLC γ 2 expression in cells stimulated with LPS, it is a very small change and does not convincingly support the authors conclusions. Perhaps more important is whether TLR4 activity alters PLC γ 2 activity, can the authors measure this?

5. Throughout the study the authors use small molecule inhibitors to measure the role of specific factors in mediating TLR4 endocytosis. How confident are the authors of the selectivity of these compounds. For example, although BAY 11-7082 has been described as an E2 inhibitor it has also been reported to be a phosphatase inhibitor as well as an IKK inhibitor (PMID:23578302). Given that the authors data overturns previous findings based on off-target effects of a small molecule (Dynasore) this is particularly pertinent.

Referee #2:

The authors have done a very extensive and unparalleled study that identifies independently operating TLR4 signaling modes that control TLR4 endocytosis and TLR4 signaling. To do this they have used a combination of inhibitors, agonists and TLR4 TIR mutants.

The manuscript submitted contains 102 references and 31000 words without spaces, 8 main figures and 10 supplementary figures.

One of the key findings in this manuscript where CD14 is suggested to shuttle LPS into the cell independent of TLR4 activity and lipid raft integrity and that that TLR4 signaling arises by parallel activation of distinct pools of TLR4 at distinct subcellular locations when signaling from the surface and from endosomes. This is an unique finding and in line with previous published papers describing that the cellular pool of TLR4 was found to be intracellular and may reach intracellular compartments without going via the plasma membrane (Hornef et al J Ex Med, 2003; Husebye et al Immunity 2010).

The manucript is exellent and I can't find much to change.

Major points:

- The authors have identified that both E1 and E2-ligases-and an undefined E3 ligase, are involved in LPS stimulated TLR4 endocytosis. However, it is intriguing that neither of the TIR -Lysine-mutants affect LPS stimulated endocytosis.
- I would like the authors to include ELISA's of TNF and CXCL10 in addition to IL-6. In addition, an immunoblot showing the expression levels of the TLR4 lysine mutants before and during LPS stimulation to rule out that the level of expression could have influenced the observed results. In addition this will show if some of the mutant fails to be degraded preferentially by lysosomal degradation.
- Mal-GFP and TRAM-GFP are mispositioned in FILIPIN treated samples: FILIPIN cause Mal-tubulation instead of endosomal location and TRAM accumulates on PM/philopodia. In both cases the enlarged endosomal structures seen in DMSO treated and LPS stimulated cells, seem to disappear. Are these phenotypes something that could be quantified? Quantification of EEA1 overlap with Mal- and TRAM-GFP.

Minor points:

278 "Similarly, while the PI3K p110 δ - selective inhibitor CAL-101 did not affect TLR4 endocytosis (Figure S10E), it inhibited LPS-induced CD14 endocytosis and Ifnb1 expression to some extent (Figure S10F, S10G).

- Can you say this when just comparing +/- LPS, is it possible to get significance when comparing DMSO to U73122 or CAL-101 treated?
- Figure S1A and S1B digram with are larger than the rest. Also in the Figure S4 the diagram and column with should be approximately the same. This is also the case for Figure 2C and 2D, S4H, S7, S10D, S10F and S10H

Referee #3:

In this manuscript, Schultz and colleagues reassessed the connection between TLR4 signalling and its endocytosis, leading to several conclusions that challenge the current view on this process. Indeed, the current predominant model proposes that, upon activation, TLR4 signals first from the plasma membrane (PM) leading to early NF κ B activation via MAL and MyD88 adaptors, followed by internalisation by endocytosis to the endosomal compartment and activation of IRF and late NF κ B through TRAM and TRIF. In contrast, the authors propose here that endocytosis of PM TLR4 is not required to trigger endosomal signalling, suggesting that this is induced by a TLR4 pool already present in endosomes. Moreover, while several studies (such as Zanoni et al Cell 2011, Tan et al Immunity 2015) indicate that TLR4 endocytosis does not require the intracellular TIR domain, this study concludes that this domain and TLR4 signalling are essential to trigger internalisation. Lastly the authors investigate the requirement for TLR4 endocytosis of the ubiquitination pathway as well as the previously proposed factors Syk, PI3K p110 and PLC γ 2.

Altogether, the questions addressed and the findings of this study are of clear interest for the field, but I think that some of the main claims need further supporting evidence, in particular as several conclusions are in contradiction with previously published data. The conclusion that TLR4 endocytosis requires TLR4 activation is supported by multiple evidence (both genetic and inhibitor based) which are overall convincing, though the reason for the discrepancy with previous studies remains unclear. In contrast, the claim that TLR4 endocytosis is not required for endosomal TLR4 signaling as well as the conclusion on the role of SYK and, in particular, PLC γ 2 need further support in my opinion (as detailed here below).

The main concern I have is the fact that the authors based many of their conclusions on the use of various chemical inhibitors, which led sometimes to contrasting results (as in the case of Syk inhibitors) or may result in off-target effects (as proposed by the authors in the case of dynasore used in previous studies). While I understand that previous studies on this topic extensively used these same inhibitors, it would be important in my opinion to consolidate these data using genetic approaches (KO, KD, expression of dominant negative constructs, ...) as much as feasible (also considering the increased accessibility of gene editing approaches).

Other major points (ordered as the text and not by importance):

1. Data in Fig S3A and S3B are not convincing nor informative being based on overexpression: MAL and MyD88 does not seem to colocalize with the PM, nor TRIF and TRAM with the endosomal EEA1 marked, and no quantification is provided. When technically possible, the localisation of the adaptors should be investigated at the endogenous level, or the data should be removed. As mentioned by the authors in the "limitation of this investigation", the lack of data assessing endogenous TLR4 localisation by imaging is indeed an important limitation of the study. While I understand that this clearly goes beyond the what can be assessed during the revision process, I would like to mention that knockin mice with an endogenous tagged TLR4-HA are available at Jackson laboratories (Strain #:031824).
2. Multiple evidence support that LPS can activate endosomal TLR4 signalling without the need of endocytosis of PM TLR4 but by triggering what to authors define as "resident endosomal TLR4" (lines 163-164), but it remains unclear the origin of this "resident" pool: does this result from direct sorting from the golgi to the endosome or is this trafficking to the PM and internalized constitutive endocytosis/recycling?
3. It is unclear if the model proposed by the authors implies that both endocytosed TLR4 and "pre-stimulation endosomal-resident" TLR4 contribute to endosomal TLR4 responses (i.e. I κ B induction) by LPS/1Z105, or if the authors exclude the contribution of PM-endocytosed TLR4 to these endosomal responses. For example, PCZ seems to inhibit CD14 internalisation (2A, S4F) but not LPS-induced I κ B induction (2A, S4C): is this due to the PM-endocytosed TLR4 (which is not affected by PCZ)? In the discussion (lines 322-324), TLR4 endocytosis is proposed as a negative regulatory mechanism, but what are the evidence that exclude the possibility that endocytosed TLR4 would contribute to amplify or sustain endosomal TLR signalling?
4. Fig 4D/E and S6G-H: While I appreciate that these reconstitution experiments are performed in primary Tlr4 ko BMM, the profile in fig S6H suggest that very few cells were analysed, likely because of low transduction efficiency in these primary cells (% of transduction efficiency is unfortunately not shown for these experiments). I think it is important to confirm these results by reconstituting TLR4 mutants in immortalized Tlr4 ko BMM, isolate cells expressing similar levels of TLR4 by FACS sorting and repeat the analysis of TLR4 internalisation and responses in these cell populations.
5. The claims related to the role of ubiquitination should be in my opinion reformulate as the mechanisms remains unclear and no clear evidence is provided that this is specific for TLR4 endocytosis. Indeed, while the authors state in lines 243-244 "We noted that E1 and E2 enzyme inhibitors also blocked LPS-induced CD14 endocytosis without impacting other cellular endocytic uptake mechanisms (Figure S9C, S9D).", this is not supported by the data, as Fig S9D show that both E2 inhibitors strongly block dextran and transferrin uptake.
6. Data and conclusions on the potential involvement of SYK, PI3K p110 and PLC γ 2 should be supported by genetic evidence, as the results with inhibitors are either not fully consistent (as for the 3 SYK inhibitors used) or their effect is relatively minor or already affecting cells before stimulation (i.e. reduction of steady state CD14 surface expression by U73122, Fig S10G).

Minor points:

- Fig S1: Please clarify the inconsistency on the effect of Dynasore on LPS-induced TLR4 internalisation between S1a (BMM: blocked) and S1d (iBMM: unaffected).
- Lines 116-118 "In the absence of TLR4 endocytosis, I κ B1 expression was not impaired but rather enhanced upon filipin or M β CD treatment (Figure 1C, S1E), whereas IL1 β expression was impaired (Figure 1D, S1E)." The comment on IL1 β is not fully

supported by the data on FIG S1E, which shows that the IL1b induction is largely unaffected in iBMM. Labelling of panels 1C and 1D are inverted in the text, as 1C shows IL1b and 1D IFN β .

- Fig S2C: Please clarify why single KO of Tram or Trif reduced IL1b induction, while this is not observed in Tram/Trif double ko in which this seems even increased.

- Fig 5D-E and lines 227-230: How the authors explain NF κ B activation independent of MAL-MyD88 and TRAM-TRIF? Can the effect of IKK-16 on endocytosis be reproduced with other NF- κ B specific inhibitors (beside MG132) or is this an off-target effect?

- Fig S10A and lines 261-262: "TLR4 lysine mutants exhibited surface expression comparable to mTLR4 WT, exhibited no defects in LPS-induced TLR4 endocytosis, and retained full signaling capacity (Figure 6F, S10A)." This is not fully correct as some reduction in IL6 production is observed.

Responses to reviewer comments - EMBOR-2024-60175V1 – Schultz et al. *TLR4 endocytosis and endosomal signaling are distinct and independent outcomes of TLR4 activation*

We would like to thank the reviewers for their very insightful comments and suggestions, which have helped us to significantly improve the manuscript and further consolidate the novel insights into how TLR4 functions are regulated.

Reviewer 1:

Overall the data is of high quality and the conclusions are mostly justified. The approach is thorough and incorporates both pharmacological inhibitors and genetic knockout cells. If anything the figures are possibly too dense and at times difficult to interpret due to the volume of data presented, particularly when the supplementary figures are included. The authors should see if there is any non-essential data that could be removed to improve the readability of the study.

We are grateful that the reviewer recognises the thorough approaches required for the high quality of data. To make the extensive new information more accessible, we have utilised Figures, Extended Figures and Supplementary Figures in the revised manuscript, while remaining highly transparent with reporting of primary data and control experiments.

1. To evaluate endocytosis the authors use a decrease in cell surface TLR4 expression as measured by flow cytometry immunofluorescence analysis. Can the authors rule out changes in total TLR4 protein expression that may contribute to decreased cell surface staining in any of their experiments?

This is a very valid point and highlights a notorious shortcoming of the field due to the lack of commercially available antibodies suitable for direct assessment of total TLR4 protein expression in the same system used for the TLR4 endocytosis studies. To address this pertinent question, we generated a TLR4-negative RAW264.7 macrophage cell line (CRISPR/Cas9) to express a Spy-tagged TLR4 construct. Application of the Spy-Catcher protein [PMID: 31052154] to these cells enabled us to specifically track stability of TLR4 that was expressed at the cell surface at the time of LPS stimulation. Our data show selective decline of the labelled cell surface-expressed higher glycosylated TLR4 upon LPS stimulation, whereas the pool of unlabelled intracellular (lower glycosylated) TLR4 is maintained or increased (the latter potentially representing replenishing of the cellular TLR4 pool after stimulation) [new Figure 1H]. Moreover, our new data highlight that the pool of surface-exposed TLR4 in macrophages represents a smaller proportion than that of intracellular TLR4, and that assessment distinguishing cell surface-exposed from intracellular TLR4 is relevant for revealing inverse dynamics of the different cellular TLR4 pools [new Figure 1H].

2. The authors state the E1 and E2 enzyme inhibitors blocked LPS-induced CD14 endocytosis without impacting other cellular endocytic uptake mechanisms - however the data shown in Figure S9D shows that the E2 inhibitor NSC697923 inhibited Dextran FITC uptake while NSC697923 and the E2 inhibitor BAY 11-7082 both inhibited Transferrin uptake. Furthermore, the Figure S9 does not contain any analysis of the effect of the E1 inhibitor PYR-41 in the same assays. Thus, the authors conclusions are not supported by the data and should be modified to accurately reflect the data.

Our data show consistent impact of the E1/E2 ubiquitination cascade inhibitors on ligand-induced TLR4 and CD14 endocytosis. Indeed, these inhibitors show a more varied impact in the control experiments that probed general endocytic pathways and we have updated the text to reflect this more clearly (lines 309-311). Of note, we prioritised the more recently developed E1 inhibitor MLN7243 (IC50: 1 nM) over PYR-41 (IC50: 10 μ M) in these control studies, as both exhibited similar effects on ligand-induced TLR4 endocytosis.

3. The authors propose that TLR4 TIR domain activity controls the cellular levels of PLC γ 2 which is also needed for TLR4 endocytosis. Do TLR4^{-/-} cells have reduced levels of PLC γ 2? which would be expected if this were the case.

We have conducted analyses in primary *Tlr4*^{-/-} macrophages and found elevated levels of total PLC γ 2 protein relative to WT macrophages [new Figure EV3H]. This is consistent with our initial observations that TAK-242 treatment of macrophages increased total PLC γ 2 protein expression [Figure EV3E-G]. It is relevant to note that inhibition of TLR4 activity by TAK-242 for only 2 hours increased PLC γ 2 expression, indicating tight control of PLC γ 2 protein expression and/or stability by tonic TLR4 signalling.

4. While the data presented in Figure 7 show statistically significant increased PLC γ 2 expression in cells stimulated with LPS, it is a very small change and does not convincingly support the authors conclusions. Perhaps more important is whether TLR4 activity alters PLC γ 2 activity, can the authors measure this?

We conducted additional experiments to probe contributions of TLR4 activity to PLC γ 2 phosphorylation (Y1217) in primary macrophages stimulated with LPS or 1Z105 over a time-course. Both TLR4 agonists induced PLC γ 2 phosphorylation alongside an elevation of total PLC γ 2 [new Figure 7B, 7C]. TAK-242 elevated total PLC γ 2 (also seen in *Tlr4*^{-/-} BMM, new Figure EV2H) and prevented ligand-induced increase of PLC γ 2 [Figure EV2E-G]. We conclude that reciprocal regulatory interactions between TLR4 and PLC γ 2 occur in macrophages.

5. Throughout the study the authors use small molecule inhibitors to measure the role of specific factors in mediating TLR4 endocytosis. How confident are the authors of the selectivity of these compounds. For example, although BAY 11-7082 has been described as an E2 inhibitor it has also been reported to be a phosphatase inhibitor as well as an IKK inhibitor (PMID:23578302). Given that the authors data overturns previous findings based on off-target effects of a small molecule (Dynasore) this is particularly pertinent.

We wholeheartedly agree with this concern, especially in light of the observations we made when comparing the impact of various endocytosis inhibitors. Throughout the study, we have invested in combining pharmacologic with genetic evidence, wherever possible, to further strengthen our main conclusions

- i) Dissociation TLR4 cell surface expression and endocytosis from endosomal TLR4 signalling.

- a. Filipin and M β CD treatment of macrophages prevents ligand-induced decline in surface TLR4, yet maintains/enhances macrophage ability for LPS- and 1Z105-induced *Ifnb1* expression [Figure 1A-C, 2C,2D].
 - b. In new experiments, we have generated macrophages with TLR4 mutated at N-glycosylation sites; N572Q and N524Q/N572Q. Macrophages expressing TLR4^{N572Q} or TLR4^{N524/572Q} lacked cell surface, but not intracellular TLR4 (as reported previously [PMID: 11706042]), yet showed *Ifnb1* expression upon LPS stimulation [new Figures 1E-G]
- ii) TLR4 activity is required for ligand-induced TLR4 endocytosis.
- a. TLR4 inhibitor TAK-242 and TLR4 TIR domain mutations in reconstituted primary cells (C745A, C745S, P712H, Δ TIR) and newly generated RAW264.7 cells (Δ TIR) prevented ligand-induced TLR4 endocytosis. (Figures 3 &4, new Figure S6K)
- iii) CD14 requirements for TLR4 endocytosis are restricted to LPS, likely reflecting functions of CD14 in LPS transfer to TLR4/MD2.
- a. The CD14-independent TLR4 agonist 1Z105 drives TLR4 endocytosis and PLC γ 2 phosphorylation (Figure 3J, new Figures 7B, 7C)
 - b. Patterns of impact of endocytic pathway inhibitors on CD14 cell surface expression in LPS and 1Z105 activated macrophages overlapped with impact on *Ifnb1* expression, but did not impact on TLR4 surface expression and *I1b* gene expression. [Figure 2]. Our approach of collective comparisons across perturbations partially overcomes the recognised limitations of pharmacological and genetic targeting of endocytic pathways [PMID: 33712737], also considering reports that TLR4 and CD14 endocytosis occurs through multiple endocytic pathways [PMID: 9820530, 16467847, 22078883].
- iv) Contributions of PLC γ 2 and SYK to TLR4 endocytosis.
- a. Small molecule PLC inhibitor U73122 and new experiments using CRISPR/Cas9-mediated knock down of PLC γ 2 impaired/prevented ligand-induced TLR4 endocytosis (Fig 7A, new Figures EV3A-D,) consistent with previous reports that PLC γ 2 positively regulates ligand-induced TLR4 endocytosis (PMID: 26106158, 22078883, 22158869).
 - b. Impaired ligand-induced TLR4 endocytosis in macrophages show in new experiments using CRISPR/Cas9-mediated knock down of SYK aligned with previous reports (PMID: 26106158, 22078883) and clarified the inconsistent results obtained with diverse small molecule inhibitors R406, BAY 61-3606, piceatannol. (Figures S10C, 10C, new Figures S10A, S10B)
- v) Contributions of ubiquitination machinery activity to TLR4 endocytosis.

- a. We base this on our observations where different, chemically distinct inhibitors that target either E1 or E2 activity (BAY 11-7082, NSC697923, PYR-41, MLN7243) exert similar effects on ligand-induced TLR4 endocytosis (Figure 6). Genetic approaches in future studies will be invaluable in mapping the relevant ubiquitination machinery and targets. With an estimated 600-1000 E3 Ub ligases and > 40 E2 enzymes (some with redundant activities), we strongly feel that such studies are beyond the scope of the current manuscript. Moreover, with 99% of cellular ubiquitination mediated by a single E1 ubiquitin-activating enzyme (Uba1) [PMID: 29334375], genetic targeting will not produce results that can be interpreted to be specific to the regulation of TLR4 endocytosis.

Reviewer 2.

The authors have identified that both E1 and E2-ligases-and an undefined E3 ligase, are involved in LPS stimulated TLR4 endocytosis. However, it is intriguing that neither of the TIR -Lysine-mutants affect LPS stimulated endocytosis.

We discuss possible explanations for these observations (lines 465-468). TLR4 has been reported to be ubiquitinated [PMID: 16467847]. If TLR4 was the direct target of ubiquitination events that regulate its endocytosis, such ubiquitination might occur on residues other than lysine. ii) a TLR4-extrinsic factor, yet to be identified, mediates ligand-induced TLR4 endocytosis. Elucidating these mechanisms will be an important future pursuit that we believe to sit outside of the scope of the current study.

I would like the authors to include ELISA's of TNF and CXCL10 in addition to IL-6. In addition, an immunoblot showing the expression levels of the TLR4 lysine mutants before and during LPS stimulation to rule out that the level of expression could have influenced the observed results. In addition this will show if some of the mutant fails to be degraded preferentially by lysosomal degradation.

As suggested, we have included ELISA analyses and demonstrate that LPS-induced CXCL10 production was sensitive to TAK-242 [new Figures 4D, S6E, EV2C]. We also attempted to assess TNF production in these samples, but TNF production was only detectable in the WT empty vector control samples. We believe this is due to a combination of factors (which influenced our original selection of IL-6 as a pro-inflammatory read-out for these experiments): i) in our experience 24 hours of TLR4 stimulation yields low TNF levels even in WT macrophages, and ii) the low transduction efficiency of the retrovirus experiments leads to TNF concentrations below the detection limit.

In new experiments, we have stably expressed WT and the lysine-null TIR-9KΔR mutant in TLR4-negative RAW264.7 macrophages. Our data show that in TIR-9KΔR TLR4 cells, the highly-glycosylated surface form of TLR4 showed elevated expression relative to WT TLR4 prior to and during LPS stimulation [new Figures EV2D, EV2E]. This suggests that while lysine residues in the TLR4 TIR domain do not contribute to ligand-induced receptor endocytosis, they may impact TLR4 trafficking and turnover.

Mal-GFP and TRAM-GFP are mispositioned in FILIPIN treated samples: FILIPIN cause Mal-tubulation instead of endosomal location and TRAM accumulates on

PM/philopodia. In both cases the enlarged endosomal structures seen in DMSO treated and LPS stimulated cells, seem to disappear. Are these phenotypes something that could be quantified? Quantification of EEA1 overlap with Mal- and TRAM-GFP.

We are grateful for these insightful comments and suggestions and agree that there might be value in future detailed analyses of TLR adapter localisation in macrophages with disrupted membrane microdomains. However, we recognise the limitations of assessing TLR adapter protein localisation (and functions) in overexpression systems (also see comments by Reviewer 3). With the additional genetic evidence (TLR4 glycosylation mutants) showing TLR4-mediated type I IFN responses in the absence of cell surface TLR4 expression, we have elected to remove this non-critical dataset so as not to distract from the main study findings.

Similarly, while the PI3K p110 δ - selective inhibitor CAL-101 did not affect TLR4 endocytosis (Figure S10E), it inhibited LPS-induced CD14 endocytosis and *Ifnb1* expression to some extent (Figure S10F, S10G). Can you say this when just comparing +/- LPS, is it possible to get significance when comparing DMSO to U73122 or CAL-101 treated?

To assess the impact of CAL-101 and U73122 on LPS-induced CD14 endocytosis, we do need to compare unstimulated to stimulated cells in the presence of either DMSO or each of the inhibitors, as presented in Figure S10I (formerly S10E). The effects of CAL-101 on LPS-induced CD14 endocytosis are small and partial, and relatively similar to that imparted on resting CD14 surface expression. We have ensured that the description of these data reflects this. Comparisons of LPS-stimulated cells across DMSO, CAL-101 and U73122 would misrepresent the data as it would disregard the impact of the inhibitors on resting cells.

Figure S1A and S1B digram with are larger than the rest. Also in the Figure S4 the diagram and column with should be approximately the same. This is also the case for Figure 2C and 2D, S4H, S7, S10D, S10F and S10H

We have adjusted the relevant figure panels, where possible.

Reviewer 3.

1. Data in Fig S3A and S3B are not convincing nor informative being based on overexpression: MAL and MyD88 does not seem to colocalize with the PM, nor TRIF and TRAM with the endosomal EEA1 marked, and no quantification is provided. When technically possible, the localisation of the adaptors should be investigated at the endogenous level, or the data should be removed. As mentioned by the authors in the "limitation of this investigation", the lack of data assessing endogenous TLR4 localisation by imaging is indeed an important limitation of the study. While I understand that this clearly go beyond the what can be assessed during the revision process, I would like to mention that knockin mice with an endogenous tagged TLR4-HA are available at Jackson laboratories (Strain #:031824).

We recognise the limitations related to TLR adapter studies in overexpressing cells and have elected to remove the dataset as suggested. (see also response to reviewer 2) With

endogenous tagging of innate immune receptors and signaling components emerging as a valuable new tool for the field (PMID: 38961291, 29368691) alongside existing mouse strains (as suggested), some of the current limitations in the field might be overcome in future studies. We believe that our new data and revised model of TLR4 regulation will provide a strong rationale and guidance for such future analyses. We agree with the Reviewer that the scope of such work exceeds the framework of this current study.

2. Multiple evidence support that LPS can activate endosomal TLR4 signalling without the need of endocytosis of PM TLR4 but by triggering what to authors define as "resident endosomal TLR4" (lines 163-164), but it remains unclear the origin of this "resident" pool: does this result from direct sorting from the golgi to the endosome or is this trafficking to the PM and internalized constitutive endocytosis/recycling?

3. It is unclear if the model proposed by the authors implies that both endocytosed TLR4 and "pre-stimulation endosomal-resident" TLR4 contribute to endosomal TLR4 responses (i.e. *Ifnb* induction) by LPS/1Z105, or if the authors exclude the contribution of PM-endocytosed TLR4 to these endosomal responses. For example, PCZ seems to inhibit CD14 internalisation (2A, S4F) but not LPS-induced *Ifnb* induction (2A, S4C): is this due to the PM-endocytosed TLR4 (which is not affected by PCZ)? In the discussion (lines 322-324), TLR4 endocytosis is proposed as a negative regulatory mechanism, but what are the evidence that exclude the possibility that endocytosed TLR4 would contribute to amplify or sustain endosomal TLR signalling?

We are summarising our responses to point 2 and 3 as we feel that these are related parts of the same larger questions.

In macrophages, TLR4 is present at the cell surface and in intracellular vesicles, including endosomal compartments. How TLR4 trafficking to distinct localisations is orchestrated is incompletely understood. In the context of our current study, we provide additional data where we selectively depleted surface-expressed TLR4 by mutating N-glycosylation sites in the TLR4 LRR domain. Consistent with previous reports (PMID: 11706042), TLR4^{N572Q} or TLR4^{N524/572Q} were not expressed on the cell surface. Yet, macrophages expressing these mutant TLR4 expressed *Ifnb1* upon LPS stimulation [new Figures 1E-G]. These observations aligned with the filipin/M β CD lipid raft inhibitor data that showed LPS-induced *Ifnb1* expression and TBK1 phosphorylation while TLR4 surface expression did not decline (Figures 1A-C, EV1D-F). Collectively, these data indicate that commonly accepted outputs of LPS-induced endosomal TLR4 signalling proceed in the absence of i) surface-expressed TLR4 and ii) ligand-induced decline of plasma membrane-expressed TLR4. This leads us to hypothesise that a pre-existing pool of signalling-competent intracellular TLR4 exists that is not reliant on constitutive or ligand-induced TLR4 endocytosis.

Our new labelling experiments in resting macrophages revealed a relatively stable cell-surface associated TLR4 population [new Figure 1H]. Nevertheless, in the context of cells expressing WT TLR4, receptor recycling might occur and feed into the intracellular pool of TLR4 in resting cells. Our new data using the SpyTag/SpyCatcher system to track TLR4 indicate that upon ligand-induced activation, cell-surface expressed TLR4 is predominantly degraded [new Figure 1H]. Whether this allows for contributions to signalling *en route* remains to be defined. The anticipated analyses required for such insights (e.g. single molecule tracking) reach beyond the scope of the current study. We have expanded the discussion to reflect our new insights (lines 423 - 439, 409 - 419).

4. Fig 4D/E and S6G-H: While I appreciate that these reconstitution experiments are performed in primary *Tlr4 ko* BMM, the profile in fig S6H suggest that very few cells were analysed, likely because of low transduction efficiency in these primary cells (% of transduction efficiency is unfortunately not shown for these experiments). I think it is important to confirm these results by reconstituting TLR4 mutants in immortalized *Tlr4 ko* BMM, isolate cells expressing similar levels of TLR4 by FACS sorting and repeat the analysis of TLR4 internalisation and responses in these cell populations.

We have conducted additional experiments with newly generated cell lines stably expressing WT and Δ TIR TLR4 in a background of TLR4-negative RAW264.7 cells. Upon LPS stimulation, cell surface expression of WT TLR4 was reduced whereas Δ TIR TLR4 expression did not decline [new Figure S6K]. Thus, these new data align with and support the observations originally made in the reconstituted primary BMM (Figure 4C,4D and S6I,S6J) that the TLR4 TIR domain is required for TLR4 endocytosis. In generating these cell lines, we noted higher cell surface expression of Δ TIR TLR4 when compared to WT TLR4 (new Figure S6K), consistent with our prior observations in reconstituted primary BMM (Figure S6I,S6J). Thus, to avoid artefacts at the extreme ends of the respective populations we did not sort for cells with TLR4 expression comparable to WT cells.

5. The claims related to the role of ubiquitination should be in my opinion reformulate as the mechanisms remains unclear and no clear evidence is provided that this is specific for TLR4 endocytosis. Indeed, while the authors state in lines 243-244 " We noted that E1 and E2 enzyme inhibitors also blocked LPS-induced CD14 endocytosis without impacting other cellular endocytic uptake mechanisms (Figure S9C, S9D).", this is not supported by the data, as Fig S9D show that both E2 inhibitors strongly block dextran and transferrin uptake.

We have revised our conclusions to more accurately reflect the observations and emphasise the need for further investigation on the potential roles of ubiquitination enzymes and targets in regulating TLR4 endocytosis (Results: Lines 309-311; Discussion: Removal of lines 352-355 from original submission).

6. Data and conclusions on the potential involvement of SYK, PI3K p110 and PLC γ 2 should be supported by genetic evidence, as the results with inhibitors are either not fully consistent (as for the 3 SYK inhibitors used) or their effect is relatively minor or already affecting cells before stimulation (i.e reduction of steady state CD14 surface expression by U73122, Fig S10G).

As suggested, we have complemented the inhibitor data with new experiments using CRISPR/Cas9-mediated depletion of SYK, PLC γ 2, or PI3K p110 δ in primary macrophages (new Figures EV3B – EV3D, S10A, S10B, S10E, S10F).

Similar to the small molecule PLC γ inhibitor U73122 (Figure EV3A), CRISPR/Cas9-mediated knock down of PLC γ 2 significantly impaired ligand-induced TLR4 endocytosis (new Figures EV3C, EV3D). The extent of the PLC γ 2 contribution in our experiments aligned with previous reports that utilised gene-deficient macrophages or the U73122 small molecule inhibitor (PMID: 26106158, 22078883). The lack of effect of the small molecule inhibitor CAL-101 (clinically used Idelalisib) on ligand-induced TLR4 endocytosis was mirrored in PI3K p110 δ -depleted cells (Fig S10G, new Figure S10E, S10F). This contrasts the previously implicated

small contribution of PI3K p110 δ in TLR4 endocytosis (PMID: 23023391). While the context of this present study did not accommodate detailed investigation of factors influencing the extent of p110 δ contributions to TLR4 endocytosis, residual p110 δ activity in inhibitor/knockdown experiments, differences between dendritic cells and macrophages, and use of kinase-dead mutants vs full knockouts might be contributing elements. We observed impaired ligand-induced TLR4 endocytosis in macrophages upon CRISPR/Cas9-mediated knock down of SYK, which aligned with previous reports (PMID: 26106158, 22078883). These new data add value as they offer some clarity on the inconsistent results obtained with the diverse small molecule inhibitors R406, BAY 61-3606 and piceatannol (new Figures S10A, S10B, Figures S10C, S10D). We believe that the comparison of the SYK inhibitors provides value to the field and have therefore retained the dataset.

Minor points:

Fig S1: Please clarify the inconsistency on the effect of Dynasore on LPS-induced TLR4 internalisation between S1a (BMM: blocked) and S1d (iBMM: unaffected).

While the observations were consistent across different independent experiments, we are unsure why there was a discrepancy in the impact of Dynasore on TLR4 endocytosis in primary BMM compared to iBMM. Our observations with dynasore inhibiting ligand-induced TLR4 endocytosis in primary mouse BMM are consistent with other reports using BMM and bone-marrow-derived dendritic cells (PMID: 18297073, 26106158, 24423728). Importantly, despite Dynasore failing to affect TLR4 endocytosis in iBMM, the LPS-induced *Ifnb1* mRNA expression was significantly reduced in these cells (Figure S1A). This further emphasises that dynasore was functional in these cells, and that TLR4 endocytosis is not a requirement for TLR4-mediated endosomal signalling.

- Lines 116-118 " In the absence of TLR4 endocytosis, *Ifnb1* expression was not impaired but rather enhanced upon filipin or M β CD treatment (Figure 1C, S1E), whereas *Il1b* expression was impaired (Figure 1D, S1E)." The comment on *IL1b* is not fully supported by the data on FIG S1E, which shows that the *IL1b* induction is largely unaffected in iBMM. Labelling of panels 1C and 1D are inverted in the text, as 1C shows *IL1b* and 1D *IFN β* .

We have provided a more nuanced description of the observations (line 133-134) and corrected the reference to panels in the text.

- Fig S2C: Please clarify why single KO of *Tram* or *Trif* reduced *IL1b* induction, while this is not observed in *Tram/Trif* double ko in which this seems even increased.

Reduced *Il1b* expression in the TRIF and TRAM single-knockout iBMMs and primary BMMs is consistent with perturbed late-stage induction of NF- κ B signalling via TRAM/TRIF/TRAF6/RIPK1 [PMID: 38599164] in these cells. We were equally surprised by the *Il1b* phenotype in the TRIF/TRAM double-knockout iBMMs. Nevertheless, this observation is consistent with observations across our study where some modes of inhibition of one axis of TLR4 signalling seemed to hyper-activate the other. For example, the TBK-1/IKKe inhibitor MRT67307 elevated *Il1b* expression (Figure S8E), the proteasome inhibitor MG132 elevated *Ifnb1* expression (Figure S9B), and filipin enhanced *Ifnb1* expression

(Figures 1C, EV1D). While this is an interesting phenomenon, insights into the underlying mechanisms are not required to support the major conclusions in the present manuscript.

Fig 5D-E and lines 227-230: How the authors explain NFkB activation independent of MAL-MyD88 and TRAM-TRIF? Can the effect of IKK-16 on endocytosis be reproduced with other NF-kB specific inhibitors (beside MG132) or is this an off-target effect?

We conducted new experiments with an IKK2 NF- κ B inhibitor (SC-514). SC-514 inhibitor did not impact LPS-induced TLR4 endocytosis, in contrast to the reproducible partial effect by IKK-16 (new Figure 5F). Divergent effects of these IKK inhibitors might reflect impact of IKK-16 on targets outside of canonical TLR4-mediated NF- κ B signalling (PMID: 22952710). We have noted these considerations in the manuscript (lines 291-292).

Fig S10A and lines 261-262: "TLR4 lysine mutants exhibited surface expression comparable to mTLR4 WT, exhibited no defects in LPS-induced TLR4 endocytosis, and retained full signaling capacity (Figure 6F, S10A)." This is not fully correct as some reduction in IL6 production is observed.

We have provided a more nuanced description of the data (line 329-332).

Dear Prof. Blumenthal,

Thank you for the submission of your revised manuscript to our editorial offices. I have now received the reports from the three referees that were asked to re-evaluate the study, you will find below. As you will see, the referees now fully support its publication in EMBO reports. Referee #3 has some suggestions to improve the manuscript, I ask you to address in a final revised manuscript. Please also provide a final p-b-p-response regarding these points.

- We now use CRediT to specify the contributions of each author in the journal submission system. CRediT replaces the author contribution section. Please use the free text box to provide more detailed descriptions and do NOT provide your final manuscript text file with an author contributions section. See also our guide to authors: <https://www.embopress.org/page/journal/14693178/authorguide#authorshippinguidelines>

- We request that primary datasets produced in a study (e.g. RNA-seq, ChIP-seq, structural and array data) are deposited in an appropriate public database. If no primary datasets have been deposited, please also state this in a dedicated section (e.g. 'No primary datasets have been generated and deposited'). Please remove any other information regarding data requests from this section.

- Please order the manuscript sections like this, using these names:

Title page - Abstract - Keywords - Introduction - Results - Discussion - Methods - Data availability section - Acknowledgements (including the funding information) - Disclosure and Competing Interests Statement - References - Figure legends - Expanded View Figure legends

- Please check again that the number "n" for how many independent experiments were performed, their nature (biological versus technical replicates), the bars and error bars (e.g. SEM, SD) and the test used to calculate p-values is indicated in the respective figure legends. Please also check that all the p-values are explained in the legend, and that these fit to those shown in the figure. Please provide statistical testing where applicable. Please avoid the phrase 'independent experiment', but clearly state if these were biological or technical replicates. Please also indicate (e.g. with n.s.) if testing was performed, but the differences are not significant. In case n=2, please show the data as separate datapoints without error bars and statistics. See also:

<http://www.embopress.org/page/journal/14693178/authorguide#statisticalanalysis>

If n<5, please show single datapoints for diagrams. Moreover:

- Please add to each legend (main, EV and Appendix figures, where applicable) a 'Data Information' section (or name the provided section like this) explaining the statistics used or providing information regarding replicates and scales. See:

- Please add the information provided in Tables S1-S4 to the Reagents & Tools table and remove them from the main manuscript text file.

- We need a proper Appendix file as pdf with all the Appendix items and their legends. Please do not upload the Appendix figures separately (and please remove these from the final submission). All the supplementary material should be supplied as a single pdf file labeled Appendix. The Appendix should have page numbers and needs to include a title page (Appendix for: ... - manuscript title; but not author names or affiliations) with a table of content on the first page (with page numbers) and legends for all content. Please follow the nomenclature Appendix Figure Sx, throughout the text, and also label the figures according to this nomenclature. Each Appendix Figure should be followed by its legend. Please update all callouts for the Appendix Figures accordingly. Finally, please remove the legends for the supplementary figures from the main manuscript text file.

- Thank you for providing the source data. Please upload the SD as one folder per figure, grouping together separate excel files for all panels for one figure (and ZIPed together) and as one folder for the EV figures (ZIPed together) and for the Appendix Figures (ZIPed together).

In addition, I would need from you uploaded separately:

- a short, two-sentence summary of the manuscript (not more than 35 words).

- two to four short (!) bullet points highlighting the key findings of your study (two lines each).

- a schematic summary figure as separate file that provides a sketch of the major findings (not a data image) in jpeg or tiff format (with the exact width of 550 pixels and a height of not more than 400 pixels) that can be used as a visual synopsis on our website.

I look forward to seeing the final revised version of your manuscript when it is ready. Please let me know if you have questions

regarding the revision.

Best,

Referee #1:

The authors have satisfactorily addressed the issues I raised in the revised manuscript.

Referee #2:

The authors have addressed all my comments satisfactory

Referee #3:

Overall, the authors sufficiently addressed my comments, and I support therefore the publication of the study. While I am aware that EMBO reports allow for a single round of revision, I enclose below few minor comments on the new data that the authors may want to address via textual changes.

Minor points :

- New figure 1H and lines 174-177: It remains unclear to me why there is an increase in the unlabelled TLR4 pool in the unstimulated control conditions. Is this a consequence of the treatment with the SpyCatcher protein? This effect should be better explained/discussed in the final version.
- New figure Ev3B-D and S10A-B and S10E-F: the generation of Syk, PLCg2 and PIK3 p110 KO significantly improved the conclusion on the impact on TLR4 endocytosis. Nevertheless, I would have appreciated to see also the effect on CD14 internalisation, to support the comments on lines 368-370.
- Repeated sentence in lines 430-433 and 434-436.

Responses to reviewer comments - EMBOR-2024-60175V2 – Schultz et al. *TLR4 endocytosis and endosomal TLR4 signaling are distinct and independent outcomes of TLR4 activation*

We were gratified by the reviewers' appreciation of the additional data and insights in response to their questions and suggestions.

Referee #1:

The authors have satisfactorily addressed the issues I raised in the revised manuscript.

Referee #2:

The authors have addressed all my comments satisfactory

Referee #3:

Overall, the authors sufficiently addressed my comments, and I support therefore the publication of the study.

While I am aware that EMBO reports allow for a single round of revision, I enclose below few minor comments on the new data that the authors may want to address via textual changes.

Minor points :

- New figure 1H and lines 174-177: It remains unclear to me why there is an increase in the unlabelled TLR4 pool in the unstimulated control conditions. Is this a consequence of the treatment with the SpyCatcher protein? This effect should be better explained/discussed in the final version.

We agree that it is not immediately clear why the unlabelled TLR4 pool would increase over time. We suspect that this is a characteristic of the SpyTag-TLR4 itself as even in resting SpyTag-TLR4 RAW264.7 cells the higher and lower molecular weight band accumulate even in the absence of SpyCatcher. To illustrate this observation, we have included new Appendix Figure S3D and reflect our assessment in lines 174-176.

- New figure Ev3B-D and S10A-B and S10E-F: the generation of Syk, PLCg2 and PIK3 p110 KO significantly improved the conclusion on the impact on TLR4 endocytosis. Nevertheless, I would have appreciated to see also the effect on CD14 internalisation, to support the comments on lines 368-370.

As the focus of our current manuscript were the mechanisms of TLR4 endocytosis, we prioritized TLR4 analyses in the CRISPR-experiments in the given time frame. We agree that the mechanisms governing CD14 endocytosis are of interest and should be expanded on in future studies. We have refined our conclusions and discussion to reflect this (lines 372-374; 484-485)

- Repeated sentence in lines 430-433 and 434-436.

The repeated sentence has been removed.

Prof. Antje Blumenthal
The University of Queensland
Australia

Dear Prof. Blumenthal,

I am very pleased to accept your manuscript for publication in the next available issue of EMBO reports. Thank you for your contribution to our journal.

Yours sincerely,
